# Parameter efficient training of deep convolutional neural networks by dynamic sparse reparameterization

## Abstract

Modern deep neural networks are highly overparameterized, and often of huge sizes. A number of post-training model compression techniques, such as distillation, pruning and quantization, can reduce the size of network parameters by a substantial fraction with little loss in performance. However, training a small network of the post-compression size *de novo* typically fails to reach the same level of accuracy achieved by compression of a large network, leading to a widely-held belief that gross overparameterization is essential to effective learning. In this work, we argue that this is not necessarily true. We describe a dynamic sparse reparameterization technique that closed the performance gap between a model compressed through iterative pruning and a model of the post-compression size trained *de novo*. We applied our method to training deep residual networks and showed that it outperformed existing reparameterization techniques, yielding the best accuracy for a given parameter budget for training. Compared to existing dynamic reparameterization methods that reallocate non-zero parameters during training, our approach achieved better performance at lower computational cost. Our method is not only of practical value for training under stringent memory constraints, but also potentially informative to theoretical understanding of generalization properties of overparameterized deep neural networks.

## 1 Introduction

Deep neural network's success is due in no small part to two of its surprising properties that starkly defy conventional wisdom. First, training of modern deep neural networks is a high-dimensional non-convex optimization problem, a kind prohibitively difficult in general from an optimization theoretic viewpoint. Yet first-order gradient-based methods, stochastic gradient descent (SGD) and its variants, prove to be very effective; training seldom gets trapped in bad local minima. Second, overparameterization (i.e. having more model parameters than training data examples) does not undermine generalization; highly overparameterized models in practice seldom overfit, a phenomenon at odds with traditional principles of statistical learning theory.

Simply put, huge models are not hard to train, and trained huge models are not bad ones.

Emerging evidence has attributed these effects to the geometry of high-dimensional loss landscapes of overparameterized deep neural networks (Dauphin et al., 2014; Choromanska et al., 2014; Goodfellow et al., 2014; Im et al., 2016; Wu et al., 2017; Liao & Poggio, 2017; Cooper, 2018; Novak et al., 2018), and to the implicit regularization properties of SGD (Brutzkus et al., 2017; Zhang et al., 2018a; Poggio et al., 2017), though a thorough theoretical understanding is not yet complete. On the other hand, practitioners, blessed in this reality, do not seem to shy away from building gigantic models. Models with parameters in the tens and hundreds of millions are common in practice, with the current record holder having a parameter count in the range of a hundred billion (Shazeer et al., 2017).

However, huge models are computationally expensive, both spatially (in terms of memory requirements) and temporally (in terms of number of operations), at both training time and inference time. Thus, scalable methods for controlling the computational complexity of deep neural networks are key to making them more useful in practice.

To achieve efficient inference, a number of well-established model compression techniques are highly effective in reducing the post-training model size with little degradation in performance. These include distillation (e.g. Bucilua et al. (2006); Hinton et al. (2015)), quantization (e.g. Hubara et al. (2016); McDonnell (2018)), low-rank decomposition (e.g. Denil et al. (2013); Jaderberg et al. (2014)), and pruning (e.g. Han et al. (2015); Zhang et al. (2018b)), to name a few.

Despite their success, post-training compression methods require the full overparameterized model to be trained in the first place. Training remains expensive. In order to make training more efficient, significant efforts have been invested into numerical innovations for training at limited precision (e.g. Courbariaux et al. (2016); Köster et al. (2017)). In contrast, however, little progress has been made in effective training at significantly reduced number of parameters.

One obvious alternative toward this goal is to seek novel network architectures, ones that are more *parameter efficient* [1]. In fact, recent architectural innovations in deep convolutional neural networks (CNNs) not only achieved better generalization performance, but parameter efficiency as well (see Section 5).

Instead of inventing new networks, an alternative approach is to achieve higher parameter efficiency directly by reparameterizing an existing model architecture. In general, any *differentiable reparameterization* can be used to augment training of a given model. Let an original network (or a layer therein) be denoted by $y = f(x; \theta)$, parameterized by $\theta \in \Theta$. Reparameterize it by $\phi \in \Phi$ through $\theta = g(\phi; \psi)$, where $g$ is differentiable w.r.t. $\phi$ but not necessarily w.r.t. $\psi$. Denote the reparameterized network by $f_\psi$, considering $\psi$ as *metaparameters* [2]:

$$y = f_\psi(x; \phi) \triangleq f(x; g(\phi; \psi)). \tag{1}$$

Training of $f_\psi$ is done by backpropagation further through $g$, as $\frac{\partial}{\partial \phi} = \frac{\partial g}{\partial \phi} \frac{\partial}{\partial g}$. If it is so chosen that $\dim(\Phi) < \dim(\Theta)$ and $f_\psi \approx f$ in terms of its generalization performance, then $f_\psi$ is a more parameter efficient approximation of $f$.

*Sparse reparameterization* is a special case where $g$ is a linear projection; $\phi$ is the non-zero entries (i.e. "weights") and $\psi$ their indices (i.e. "connectivity") in the original parameter $\theta$. Likewise, *parameter sharing* (or more generally *linear constraints of parameters*) is a similar special case of linear reparameterization where $\phi$ is the tied parameters and $\psi$ their indices. In this sense a convolution layer can be considered a reparameterization of a fully connected layer.

Further, if metaparameters $\psi$ are fixed during the course of training, the reparameterization is *static*, whereas if $\psi$ is adjusted adaptively during training, we call it *dynamic* reparameterization.

Now, the question is: given a full model $f$, is it possible to find a more efficient reparameterization $f_\psi$ that, trained *de novo*, can generalize comparably well? The success of various model compression techniques suggests that such reparameterizations might exist for most well-known models, and a recent study (Frankle & Carbin, 2018) made successful *post hoc* identifications of sparse reparameterized small networks with precisely such properties. Nevertheless, attempts at training small networks *de novo* typically yield results significantly underperforming networks obtained by compressing larger models (Zhu & Gupta, 2017). This has led to a commonly held belief that gross overparameterization is necessary for effective training. Here we argue that this is not true by presenting a dynamic sparse reparameterization technique able to train sparse models *de novo* without the need to compress a large model, a desirable feature for training on memory- and power-constrained devices.

Our contributions are listed as follows.
1. We showed that it is possible to train a small sparse network directly without a larger than inference-time parameter footprint at any stages of training, yet still achieving generalization performance at least on par with post-training iterative pruning of large dense models, yielding the most parameter efficient model at a given sparsity.
2. We showed that our method is more scalable and efficient, and leads to significantly better accuracy than existing dynamic sparse reparameterization training techniques.

---

[1] We formally introduce the notion of *parameter efficiency* as follows: given a dataset and a task, if model family $A$ achieves a specific level of generalization performance with fewer parameters than model family $B$, we say $A$ is more *parameter efficient* than $B$ at that performance level.

[2] We use the term *metaparameter* to refer to the parameters $\psi$ of the reparameterization function $g$. They differ from parameters $\phi$ in that they are not simultaneously optimized with model parameters, and they differ from hyperparameters in that they define meaningful structures of the model which are required for inference.

## 2 RELATED WORK

Training of differentiably reparameterized networks has been proposed in numerous studies before.

**Dense reparameterization** Several dense reparameterization techniques sought to reduce the size of fully connected layers. These include low-rank decomposition (Denil et al., 2013), fastfood transform (Yang et al., 2014), ACDC transform (Moczulski et al., 2015), HashedNet (Chen et al., 2015), low displacement rank (Sindhwani et al., 2015) and block-circulant matrix parameterization (Treister et al., 2018).

Note that similar reparameterizations were also used to introduce certain algebraic properties to the parameters for purposes other than reducing model sizes, e.g. to make training more stable as in unitary evolution RNNs (Arjovsky et al., 2015) and in weight normalization (Salimans & Kingma, 2016), to inject inductive biases (Thomas et al., 2018), and to alter (Dinh et al., 2017) or to measure (Li et al., 2018) properties of the loss landscape. All dense reparameterization methods to date are static.

**Sparse reparameterization** Successful training of sparse reparameterized networks usually employs iterative pruning and retraining, e.g. Han et al. (2015); Narang et al. (2017); Zhu & Gupta (2017) [3]. Training typically starts with a large pre-trained model and sparsity is gradually increased during the course of fine-tuning. Training a small, static, and sparse model *de novo* always fared much worse than training a large one to begin with (Zhu & Gupta, 2017).

Frankle & Carbin (2018) successfully identified small and sparse subnetworks post-training which, when trained in isolation, reached a similar accuracy as the enclosing big network. They further showed that these subnetworks were sensitive to initialization, and hypothesized that the role of overparameterization is to provide a large number of candidate subnetworks, thereby increasing the likelihood that one of these subnetworks will have the necessary structure and initialization needed for effective learning.

Most closely related to our work are dynamic sparse reparameterization techniques that emerged only recently. Like ours, these methods adaptively alter, by certain heuristic rules, reparameterization during training. Sparse evolutionary training (Mocanu et al., 2018) used magnitude-based pruning and random growth at the end of each training epoch. NeST (Dai et al., 2017; 2018) iteratively grew and pruned parameters and neurons during training; parameter growth was guided by gradient and pruning by magnitude. Deep rewiring (Bellec et al., 2017) combined sparse reparameterization with stochastic parameter updates for training. These methods were mostly concerned with sparsifying fully connected layers and applied to relatively small and shallow networks. We show that the method we propose in this paper is more scalable and computationally efficient than these previous approaches, while achieving better performance on deep convolutional networks.

## 3 METHODS

In this section, we describe our dynamic sparse reparameterization method. Some notations first. Let all reparameterized weight tensors in the original network be denoted by $\{\mathbf{W}_l\}$, where $l = 1, \cdots, L$ indexes layers. Let $N_l$ be the number of parameters in $\mathbf{W}_l$, and $N = \sum_l N_l$ the total parameter count.

Sparse reparameterize $\mathbf{W}_l = g(\phi_l; \psi_l)$, where function $g$ places components of parameter $\phi_l$ into positions in $\mathbf{W}_l$ indexed by $\psi_l \in \mathbf{\Psi}_{M_l}(\{1, \cdots, N_l\})$ [4], s.t. $W_{l,\psi_{l,i}} = \phi_{l,i}, \forall i$ indexing components. Let $M_l < N_l$ be the dimensionality of $\phi_l$ and $\psi_l$, i.e. the number of non-zero weights in $\mathbf{W}_l$. Define $s_l = 1 - \frac{M_l}{N_l}$ as the *sparsity* of $\mathbf{W}_l$. Global sparsity is then defined as $s = 1 - \frac{M}{N}$ where $M = \sum_l M_l$.

During the whole course of training, we kept global sparsity constant, specified by hyperparameter $s \in (0, 1)$. Reparameterization was initialized by uniformly sampling positions in each weight tensor

---

[3] Note that these, as well as all other techniques we benchmark against in this paper, impose *non-structured* sparsification on parameter tensors, yielding *sparse* models. There also exist a class of *structured* pruning methods that "sparsify" at channel or layer granularity, e.g. Luo et al. (2017) and Huang & Wang (2017), generating essentially small *dense* models. We describe a full landscape of existing methods in Appendix C.

[4] By $\mathbf{\Psi}_p(Q) \triangleq \{\sigma(\Psi) : \Psi \in 2^Q, |\Psi| = p, \sigma \in S_p\}$ we denote the set of all cardinality $p$ ordered subsets of finite set $Q$.

**Algorithm 1:** Reallocate free parameters within and across weight tensors

**Input:** $\left\{\left(\boldsymbol{\phi}_l^{(t)}, \boldsymbol{\psi}_l^{(t)}\right)\right\}, M^{(t)}, H^{(t)}$            ▷ From step $t$

**Output:** $\left\{\left(\boldsymbol{\phi}_l^{(t+1)}, \boldsymbol{\psi}_l^{(t+1)}\right)\right\}, M^{(t+1)}, H^{(t+1)}$        ▷ To step $t+1$

**Need:** $K, \delta$        ▷ Target number of parameters to be pruned and its fractional tolerance

1   **for** $l \in \{1, \cdots, L\}$ **do**        ▷ For each reparameterized weight tensor

2     $\Pi_l^{(t)} \leftarrow \left\{i : |\phi_{l,i}^{(t)}| < H^{(t)}\right\}$     ▷ Indices of subthreshold components of $\phi_l^{(t)}$ to be pruned

3     $\left(K_l^{(t)}, R_l^{(t)}\right) \leftarrow \left(|\Pi_l^{(t)}|, M_l^{(t)} - |\Pi_l^{(t)}|\right)$     ▷ Numbers of pruned and surviving weights

4   **if** $\sum_l K_l^{(t)} < (1 - \delta)K$ **then**        ▷ Too few parameters pruned

5     $H^{(t+1)} \leftarrow 2H^{(t)}$        ▷ Increase pruning threshold

6   **else if** $\sum_l K_l^{(t)} > (1 + \delta)K$ **then**        ▷ Too many parameters pruned

7     $H^{(t+1)} \leftarrow \frac{1}{2}H^{(t)}$        ▷ Decrease pruning threshold

8   **else**        ▷ A proper number of parameters pruned

9     $H^{(t+1)} \leftarrow H^{(t)}$        ▷ Maintain pruning threshold

10 **for** $l \in \{1, \cdots, L\}$ **do**        ▷ For each reparameterized weight tensor

11     $G_l^{(t)} \leftarrow \left\lfloor \frac{R_l^{(t)}}{\sum_l R_l^{(t)}} \sum_l K_l^{(t)} \right\rfloor$        ▷ Redistribute parameters for growth

12     $\tilde{\boldsymbol{\psi}}_l^{(t)} \sim \mathcal{U}\left[\boldsymbol{\Psi}_{G_l^{(t)}}\left(\{1, \cdots, N_l\} \setminus \left\{\psi_{l,i}^{(t)}\right\}\right)\right]$     ▷ Sample zero positions to grow new weights

13     $M_l^{(t+1)} \leftarrow M_l^{(t)} - K_l^{(t)} + G_l^{(t)}$        ▷ New parameter count

14     $\left(\boldsymbol{\phi}_l^{(t+1)}, \boldsymbol{\psi}_l^{(t+1)}\right) \leftarrow \left(\left[\boldsymbol{\phi}_{l,i \notin \Pi_l^{(t)}}^{(t)}, \mathbf{0}\right], \left[\boldsymbol{\psi}_{l,i \notin \Pi_l^{(t)}}^{(t)}, \tilde{\boldsymbol{\psi}}_l^{(t)}\right]\right)$     ▷ New reparameterization

at the global sparsity $s$, i.e. $\boldsymbol{\psi}_l^{(0)} \sim \mathcal{U}\left[\boldsymbol{\Psi}_{M_l^{(0)}}\left(\{1, \cdots, N_l\}\right)\right], \forall l$, where $M_l^{(0)} = \lfloor(1 - s)N_l\rfloor$. Associated parameters $\boldsymbol{\phi}_l^{(0)}$ were randomly initialized.

Dynamic reparameterization was done periodically by repeating the following steps during training:

1. Train the model (currently reparameterized by $\left\{\left(\boldsymbol{\phi}_l^{(t)}, \boldsymbol{\psi}_l^{(t)}\right)\right\}$) for $P$ batch iterations;
2. Reallocate free parameters within and across weight tensors following Algorithm 1 to arrive at new reparameterization $\left\{\left(\boldsymbol{\phi}_l^{(t+1)}, \boldsymbol{\psi}_l^{(t+1)}\right)\right\}$.

The adaptive reallocation is in essence a two-step procedure: a global pruning followed by a tensor-wise growth. Specifically our algorithm has the following key features:

- Pruning was based on magnitude of weights, by comparing all parameters to a global threshold $H$, making the algorithm much more scalable than methods relying on layer-specific pruning.
- We made $H$ adaptive, subject to a simple setpoint control dynamics that ensured roughly $K$ weights to be pruned globally per iteration. This proved to be much cheaper computationally than pruning exactly $K$ smallest weights, which requires sorting all weights in the network.
- Growth was by uniformly sampling zero weights and tensor-specific, thereby achieving a reallocation of parameters across layers. The heuristic guiding growth is

$$G_l^{(t)} = \left\lfloor \frac{R_l^{(t)}}{\sum_l R_l^{(t)}} \sum_l K_l^{(t)} \right\rfloor, \tag{2}$$

where $K_l^{(t)}$ and $R_l^{(t)} = M_l^{(t)} - K_l^{(t)}$ are the pruned and surviving parameter counts, respectively. This rule allocated more free parameters to weight tensors with more surviving entries, while keeping the global sparsity the same by balancing numbers of parameters pruned and grown [5].

---

[5] Note that an exact match is not guanranteed due to rounding errors in Eq. 2 and the possibility that $M_l^{(t)} - K_l^{(t)} + G_l^{(t)} > N_l$, i.e. free parameters in a weight tensor exceeding its dense size after reallocation. We added an extra step to redistribute parameters randomly to other tensors in these cases, thereby assuring an exact global sparsity.

**Table 1:** Datasets and models used in experiments

| Dataset | MNIST | CIFAR10 | Imagenet |
|---|---|---|---|
| Model | LeNet-300-100 (LeCun et al., 1998) | WRN-28-2 (Zagoruyko & Komodakis, 2016) | Resnet-50 (He et al., 2015) |
| Architecture | F300
F100
F10 | C16/3×3
[C16/3×3,C16/3×3]×4
[C64/3×3,C64/3×3]×4
[C128/3×3,C128/3×3]×4
GlobalAvgPool, F10 | C64/7×7-2, MaxPool/3×3-2
[C64/1×1, C64/3×3, C256/1×1]×3
[C128/1×1, C128/3×3, C512/1×1]×4
[C256/1×1, C256/3×3, C1024/1×1]×6
[C512/1×1, C512/3×3, C2048/1×1]×3
GlobalAvgPool, F1000 |
| # Parameters | 267K | 1.5M | 25.6M |

For brevity architecture specifications omit batch normalization and activations. Fully connected (F) and convolutional (C) layers are specified with output size and kernel size, Max pooling (MaxPool) with kernel size and none with global average pooling (GlobalAvgPool). Brackets enclose residual blocks postfixed with repetition numbers; downsampling convolution in the first block of a scale group is implied.

The entire procedure can be fully specified by hyperparameters $\left(s, P, K, \delta, H^{(0)}\right)$. For specific values used in experiments see implementational details in Appendix A.

## 4 EXPERIMENTAL RESULTS

We evaluated our method in three sets of experiments (Table 1). We chose more modern convolutional networks, Resnet (He et al., 2015) and Wide Resnet (Zagoruyko & Komodakis, 2016), with parameter efficiency superior to earlier models, e.g. AlexNet (Krizhevsky et al., 2012) and VGG (Simonyan & Zisserman, 2014), thanks to the adoption of skip connections and use of global average pooling over fully connected layers. Such a setup makes a much stronger case for our method because compression is much harder for these recent models. Pre-activation batch normalization was used in all cases.

Dynamic sparse reparameterization was applied to all weight tensors of fully connected and convolutional layers (with the exception of downsampling convolutions and the first convolutional layer taking the input image), while all biases and parameters of normalization layers were kept dense [6].

At a specific sparsity $s$, we compared our method (*dynamic sparse*) against six baselines:
- *Full dense*: original large and dense model, with $N$ parameters;
- *Thin dense*: original model with less wide layers, such that it had $(1 - s)N$ parameters;
- *Static sparse*: original model reparameterized to sparsity $s$, then trained with connectivity fixed;
- *Compressed sparse*: state-of-the-art compression of the original model by iterative pruning and retraining the original model to target sparsity $s$ (Zhu & Gupta, 2017);
- *DeepR*: sparse model trained by using Deep Rewiring (Bellec et al., 2017);
- *SET*: sparse model trained by using Sparse Evolutionary Training (Mocanu et al., 2018).

Note that *compressed sparse* is a compression method that starts training with a dense model, whereas *DeepR* and *SET*, like ours, are dynamic reparameterization techniques that maintain sparsity throughout training. See Appendix A for hyperparameters used in the experiments.

**LeNet-300-100 on MNIST** Like previous work on post-training compression, e.g. Han et al. (2015), and dynamic reparameterization, e.g. Bellec et al. (2017), we first experimented with a simple LeNet-300-100 trained on MNIST. We found that *static sparse* grossly underperformed *compressed sparse*, a gap effectively closed by our *dynamic sparse* method (Figure 1a) with the following nuances. *Dynamic sparse* slightly outperformed *compressed sparse* at very high sparsity (i.e. low parameter count), a trend reversed, albeit weakly, at lower sparsities.

A breakdown of post-training sparsities across layers of the network (Figure 1b) reveals reliable patterns emerged from the dynamics of our adaptive reparameterization algorithm. This is consistent

---

[6] As described in Section 3, global sparsity $s$ was defined as the overall sparsity of all reparameterized parameter tensors, not the entire model, which had a tiny fraction of dense parameters in our examples.

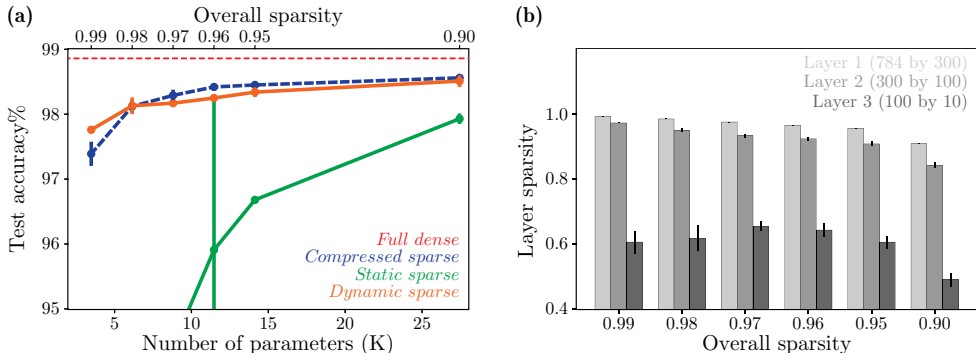

**Figure 1:** LeNet-300-100 on MNIST. (a) Test accuracy plotted against number of trainable parameters for different methods. Dashed lines are used for full dense model and for compression methods, whereas all reparameterization methods maintaining a fixed sparsity level throughout training are represented by solid lines. Circular symbols mark the median of 5 runs, error bars standard deviation. Parameter counts include all trainable parameters, i.e. reparameterized sparse paremeter tensors plus all other parameter tensors that were kept dense, such as those of batch normalization layers. For small numbers of trainable parameters, the static sparse model fails to learn (not shown, out of ordinate range). (b) Layer-wise breakdown of final sparsities emerged from our dynamic sparse parameterization algorithm (Algorithm 1) at different levels of overall sparsity. Mean and standard deviation from 5 runs.

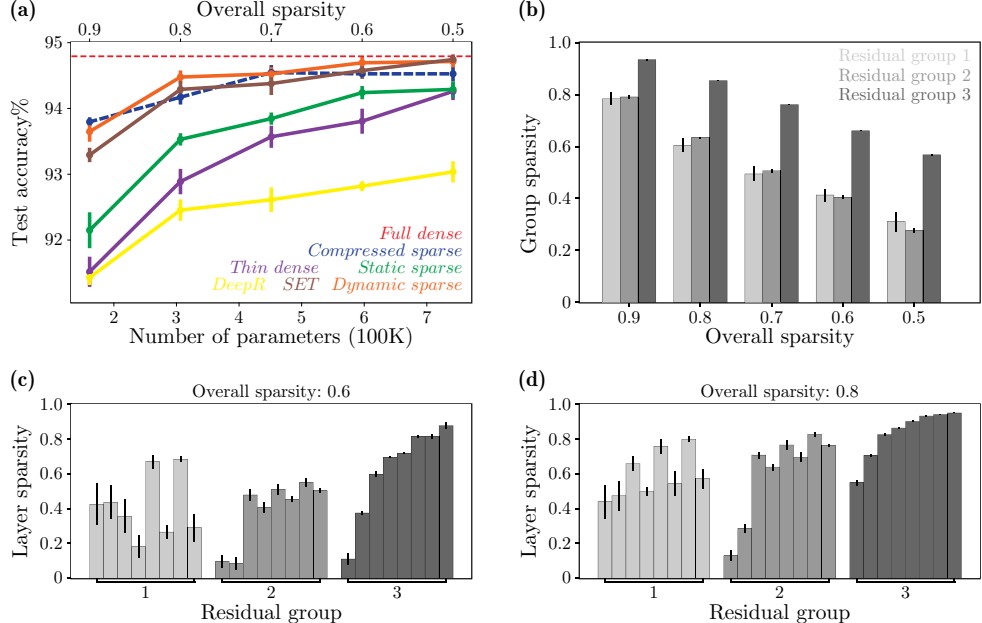

**Figure 2:** WRN-28-2 on CIFAR10. (a) Test accuracy plotted against number of trainable parameters for different methods. Conventions same as in Figure 1a. (b) Breakdown of final sparsities emerged from our dynamic sparse parameterization algorithm (Algorithm 1) at different levels of global sparsity. (c) A further breakdown of final sparsities of individual residual blocks in the WRN groups at three scales, at the overall sparsity of 0.6. (d) Same as (c) at overall sparsity of 0.8. All results in (b, c & d) are mean and standard deviation from 5 runs.

with previous observations (Bellec et al., 2017) that, given a fixed global sparsity, it was wiser to allocate more free parameters to the last layer, than to impose constant sparsity on all layers.

**WRN-28-2 on CIFAR10** Next, we experimented with a Wide Resnet model WRN-28-2 (Zagoruyko & Komodakis, 2016) trained to classify CIFAR10 images (see Appendix A for details of implementation). As shown in Figure 2a, *static sparse* and *thin dense* significantly underperformed the state-of-the-art *compressed sparse* model, whereas our method *dynamic sparse* significantly outperformed it. Deep rewiring significantly lagged all other method. While the performance of SET

**Table 2:** Test accuracy% (top-1, top-5) of Resnet-50 trained on Imagenet

| Final overall sparsity (# Parameters) | | | 0.8 (7.3M) | | 0.9 (5.1M) | | 0.0 (25.6M) | |
|---|---|---|---|---|---|---|---|---|
| Reparameterization | static | *Thin dense* | 71.6 [-3.3] | 90.3 [-2.1] | 69.4 [-5.5] | 89.2 [-3.2] | | |
| | | *Static sparse* | 70.4 [-4.5] | 89.8 [-2.6] | 66.4 [-8.5] | 87.4 [-5.0] | | |
| | dynamic | *DeepR* (Bellec et al., 2017) | - [-] | - [-] | - [-] | - [-] | 74.9 [0.0] | 92.4 [0.0] |
| | | *SET* (Mocanu et al., 2018) | 72.6 [-2.3] | 91.2 [-1.2] | 70.4 [-4.5] | 90.1 [-2.3] | | |
| | | *Dynamic sparse* (Ours) | **73.3** [**-1.6**] | **92.4** [**0.0**] | **71.6** [**-3.3**] | **90.5** [**-1.9**] | | |
| Compression | | *Compressed sparse* (Zhu & Gupta, 2017) | 73.2 [-1.7] | 91.5 [-0.9] | 70.3 [-4.6] | 90.0 [-2.4] | | |
| | | *ThiNet* (Luo et al., 2017) | *68.4* [*-4.5*] | *88.3* [*-2.8*] | (at 8.7M parameter count) | | | |
| | | *SSS* (Huang & Wang, 2017) | *71.8* [*-4.3*] | *90.8* [*-2.1*] | (at 15.6M parameter count) | | | |

Numbers in square brackets are differences from the *full dense* baseline. Romanized numbers are results of our experiments, and italicized ones taken directly from the original paper. Performance of two structured pruning methods, ThiNet and Sparse Structure Selection (SSS), are also listed for comparison (below the double line, see Appendix C for discussion of their relevance); note the difference in parameter counts.

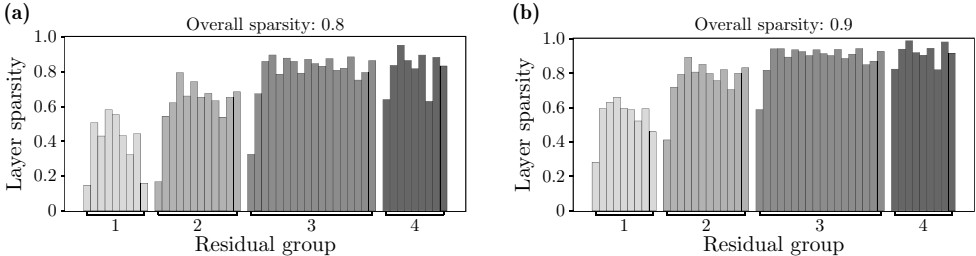

**Figure 3:** Block-wise breakdown of final sparsities of Resnet-50 trained on Imagenet. (a) At overall sparsity 0.8. Similar to Figure 2c. (b) Same as (a) at overall sparsity of 0.9.

was on par with *compressed sparse*, it lagged behind *dynamic sparse* at high sparsity levels. At low sparsity levels SET largely closed the gap to *compressed sparse*.

Again, consistent sparsity patterns emerged from the adaptive reparameterization dynamics of Algorithm 1 (Figure 2b, c & d). We observed two rough trends: (a) larger parameter tensors tended to be sparser than smaller ones, and (b) deeper layers tended to be sparser than shallower ones.

**Resnet-50 on Imagenet** Finally, we experimented with the Resnet-50 bottleneck architecture (He et al., 2015) trained on Imagenet (see Appendix A for details of implementation). We tested two global sparsity levels, 0.8 and 0.9 (Table 2). Again, our method (*dynamic sparse*) outperformed state-of-the-art sparse compression (*compressed sparse*), which in turn outperformed *static sparse* and *thin dense* baselines. We also list in Table 2 two representative methods of *structured* pruning (see Appendix C), ThiNet (Luo et al., 2017) and Sparse Structure Selection (Huang & Wang, 2017), which, consistent with recent criticisms (Liu et al., 2018), underperformed static dense baselines.

Consistent with the aforementioned experiments with LeNet-100-10 and WRN-28-2, reliable sparsity patterns across layers emerged from dynamic parameter reallocation during training, displaying the same empirical trends described above (Figure 3).

**Computational overhead of dynamic reparameterization** We assessed the additional computational cost incurred by reparameterization steps (Algorithm 1) during training, and compared ours

**Table 3:** Computational overhead of dynamic reparameterization during training

|  | WRN-28-2 on CIFAR10 | Resnet-50 on Imagenet |
| --- | --- | --- |
| *DeepR* (Bellec et al., 2017) | $4.466 \pm 0.358$ | $5.636 \pm 0.218$ |
| *SET* (Mocanu et al., 2018) | $1.087 \pm 0.049$ | $1.009 \pm 0.002$ |
| *Dynamic sparse* (Ours) | $\mathbf{1.083 \pm 0.051}$ | $\mathbf{1.005 \pm 0.004}$ |

Shown are median ratios of wall-clock epoch times for training *with* over *without* reparameterization, standard deviation estimated from 25 epochs. WRN-28-2 on CIFAR10 was trained on a single Nvidia Titan Xp GPU, and Resnet-50 on Imagenet on four with data parallelism (also see Appendix A for implementation details).

with existing dynamic sparse reparameterization techniques, DeepR and SET (Table 3). Because both SET and ours reallocate parameters only intermittently (every few hundred training iterations), the computational overhead was negligible for the experiments presented here[7]. DeepR, however, requires adding noise to gradient updates as well as reallocating parameters every training iteration, and therefore led to a significantly larger overhead.

## 5    DISCUSSION

In this work, we described and validated the best known solution so far to the following problem: given a small, fixed budget of parameters for a deep CNN throughout training time, how to train it to yield the best generalization performance. Our method used a dynamic reparameterization technique that adaptively reallocated free parameters across the network based on a simple heuristic. We demonstrated that this method not only fared much better than static reparameterization techniques, but also significantly outperformed even the state-of-the-art sparse compression methods. Note that compression is a much more forgiving situation where the parameter budget is not imposed during the entire course, but only at the end, of training.

Thus, our method yielded *the most parameter efficient sparse reparameterization* of a given CNN. The past few years have seen a steady improvement in parameter efficiency of state-of-the-art CNN models thanks to innovations in network architectures. Specifically, the introduction of residual modules and the use of global pooling in place of fully connected layers are largely responsible, as convolutional layers are more parameter efficient than fully connected layers in general. For example, AlexNet (Krizhevsky et al., 2012) had a majority of its parameters in fully connected layers, and this fraction has been decreasing in VGG (Simonyan & Zisserman, 2014), Inception (Szegedy et al., 2014), and more recently in Resnets (He et al., 2015) it was reduced to close to none. Meanwhile, compact architectures with far fewer parameters also emerged at reasonable costs of performance degradation, e.g. SqueezeNet (Iandola et al., 2016) and MobileNet (Howard et al., 2017), for niche use cases where computing resources are limited.

Here we took a different, but complementary, approach. Instead of searching the space of network architectures for new models with superior parameter efficiency, we sought to reparameterize a given model to make it more parameter efficient. Such techniques have a history as long as modern convolutional networks. Most early methods used dense reparameterization (Denil et al., 2013; Yang et al., 2014; Moczulski et al., 2015; Sindhwani et al., 2015), effective in reducing redundancy in fully connected layers of big sizes, such as in AlexNet. These became less relevant as models became increasingly free of large linear layers but heavy in convolutional ones [8].

Sparse reparameterization, on the other hand, has been shown useful in compression of state-of-the-art deep CNNs (Han et al., 2015; Narang et al., 2017). However, a prominent empirical observation from these compression studies is that training a small sparse model *de novo* could never reach the same accuracy achieved by one of the same size compressed down from a large model (Zhu & Gupta, 2017). Why does it seem impossible to train a compact network but easy to compress a large one?

---

[7] Because of the rather negligible overhead, the reduced operation count thanks to the elimination of sorting operations did not amount to a substantial improvement over SET on GPUs. However, our method has more prominent strengths over SET in its abilities to produce better sparse models and to reallocate free parameters automatically (see Appendix C).

[8] Note that a dense reparameterization trick, HashedNet (Chen et al., 2015), is closely related to sparse methods and can be readily applied to CNNs. We compared our method with HashedNet in Appendix B.

The "lottery ticket hypothesis" (Frankle & Carbin, 2018) offered a plausible explanation. The authors identified, by post-training pruning, trainable sparse models that, albeit sensitive to initialization, generalized comparably well (called "winning tickets"). They further posited that, the necessity of starting training with a large model is because only a big network with combinatorially a huge number of small subnetworks has a high probability of "winning the initialization lottery". Similarly, here we argue that *post hoc* identification of trainable compact (not necessarily sparse) reparameterization is actually rather trivial also because *the optimization trajectory is typically a very low-dimensional object*. Take for example the training of Resnet-152 (parameter count 60.2M) as described by He et al. (2015): parameter updates for 600K iterations reliably send a random initialization to a state-of-the-art solution. One could simply take a trivial *post hoc* reparameterization as a linear projection onto the span of the 600K parameter updates which, even if they are all linearly independent from each other, amount to less than 1% of the total parameter count. The low dimensionality of optimization trajectory is consistent with emerging evidence suggesting that optima of overparameterized deep neural networks are usually high-dimensional manifolds with low co-dimensions (Cooper, 2018), and that such optima have superior generalization properties (Wu et al., 2017) and favored by SGD (Zhang et al., 2018a; Poggio et al., 2017).

What prohibits any *post hoc* reparameterized compact model from being effectively trained *de novo* is however its sensitivity to initialization–though optimization trajectories are each low-dimensional, their linear spans do not necessarily overlap when starting from different initial points. An obvious way "to win the lottery" is to purchase all available tickets (i.e. having a huge model to cover all possible linear spans of optimization trajectories), but it is also conceivably feasible "to cheat" by adaptively changing the number on a single ticket as the lottery result is being announced (i.e. dynamic reparameterization that continually re-orients a low-dimensional parameter manifold tangentially along an optimization trajectory). We believe our method did exactly this.

We were inspired by previous methods of dynamic sparse reparameterization, e.g. DeepR (Bellec et al., 2017) and SET (Mocanu et al., 2018) (also see Appendix C). Compared to these techniques, our method has a number of advantages. First, ours produced better performing sparse CNNs, fully closed the gap to compression of large dense models. Second, our reparameterization incurred the least computational overhead during training by infrequent and cheap parameter reallocation. Finally, our method is highly scalable thanks to its automatic reallocation of parameters across layers, without the need of manually configuring sparsity for each layer [9].

A noteworthy observation from our experiments is that the highest parameter efficiency was achieved by *non-structured* sparsification (see Appendix C for details). Accuracy suffered significantly when structure was imposed on the sparsity patterns (see Appendix D). Because computations involving fine-grained sparsity cannot be readily accelerated on GPUs, a number of compression methods by *structured pruning* (see Appendix C) remove entire channels from deep CNNs to yield smaller networks with dense parameter tensors. However, many of these structured pruning methods produced compressed models no better than direct training of a *thin dense* model (Liu et al., 2018) (also see Table 2 for examples compared with our method).

Our findings suggest that it is possible to train sparse models directly to reach generalization performances equal to or better than those of dense or sparse networks of comparable size produced by compression. This warrants a renewed look at the practical choices of best hardware architectures for training, between computing devices optimized for dense linear algebra versus those suited for sparse operations.

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

## APPENDIX A   DETAILS OF IMPLEMENTATION

We implemented all models and reparameterization mechanisms using `pytorch`. Experiments were run on GPUs, and all sparse tensors were represented as dense tensors filtered by a binary mask [10]. Source code to reproduce all experiments is available on GitHub:

```
Link suppressed for the sake of anonymity during review process.
```

**Training**   Hyperparameter settings for training are listed in the first block of Table 4.  Standard mild data augmentation was used in all experiments for CIFAR10 (random translation, cropping and horizontal flipping) and for Imagenet (random cropping and horizontal flipping).

**Sparse compression baseline**   We compared our method against iterative pruning methods (Han et al., 2015; Zhu & Gupta, 2017). We start from a full dense model trained with hyperparameters provided in the first block of Table 4 and then gradually prune the network to a target sparsity in $T$ steps. As in Zhu & Gupta (2017), the pruning schedule we used was

$$s^{(t)} = s + (1 - s) \left( 1 - \frac{t}{T} \right)^3 , \tag{3}$$

where $t = 0, 1, \cdots, T$ indexes pruning steps, and $s$ the target sparsity reached at the end of training. Thus, this baseline (labeled as *compressed sparse* in the paper) was effectively trained for more iterations (original training phase plus compression phase) than our *dynamic sparse* method, to the best of our knowledge a so far strongest baseline to benchmark our method.

Hyperparameter settings for sparse compression are listed in the second block of Table 4.

**Dynamic reparameterization (ours)**   Hyperparameter settings for dynamic sparse reparameterization (Algorithm 1) are listed in the third block of Table 4.

**Sparse Evolutionary Training (SET)**   Because the larger-scale experiments here (WRN-28-2 on CIFAR10 and Resnet-50 on Imagenet) were not attempted by Mocanu et al. (2018), no specific settings for reparameterization in these cases were available in the original paper. In order to make a fair comparison, we used the same hyperparameters as those used in our dynamic reparameterization scheme (third block in Table 4). At each reparameterization step, the weights in each layer were sorted by magnitude and the smallest fraction was pruned. An equal number of parameters were then randomly allocated in the same layer and initialized to zero. For control, the total number of reallocated weights at each step was chosen to be the same as our dynamic reparameterization method, as was the schedule for reparameterization.

**Deep Rewiring (DeepR)**   The fourth block in Table 4 contain hyperparameters for the DeepR experiments. We refer the reader to Bellec et al. (2017) for details of the deep rewiring algorithm and for explanation of the hyperparameters.

**Exceptions**   For reasons described below, we made a few minor exceptions in sparsification for certain model layers in our experiments.
- The sparsity of the last linear layer in LeNet-300-100 was allowed to be sparsified to at most 90% sparsity. In cases where global sparsity was over 90%, we redistribute free parameters from earlier layers to maintain the total parameter count. This is because, if the last layer was allowed to be overly sparse, performance sustained huge degradation unsuited for meaningful comparison.
- The last linear layer of WRN-28-2 was always kept dense. It has a negligible fraction of parameter count.

---

[10] This is a mere implementational choice for ease of experimentation given available hardware and software, which did not save memory because of sparsity. With computing substrate optimized for sparse linear algebra, our method is duly expected to realize the promised memory efficiency.

**Table 4:** Hyperparameters for all experiments presented in the paper

| Experiment | LeNet-300-100 on MNIST | WRN-28-2 on CIFAR10 | Resnet-50 on Imagenet |
|---|---|---|---|
| **Hyperparameters for training** | | | |
| Number of training epochs | 100 | 200 | 100 |
| Mini-batch size | 100 | 100 | 256 |
| Learning rate schedule (epoch range: learning rate) | 1 - 25: 0.100
26 - 50: 0.020
51 - 75: 0.040
76 - 100: 0.008 | 1 - 60: 0.100
61 - 120: 0.020
121 - 160: 0.040
161 - 200: 0.008 | 1 - 30: 0.1000
31 - 60: 0.0100
61 - 90: 0.0010
91 - 100: 0.0001 |
| Momentum (Nesterov) | 0.9 | 0.9 | 0.9 |
| $L^1$ regularization multiplier | 0.0001 | 0.0 | 0.0 |
| $L^2$ regularization multiplier | 0.0 | 0.0005 | 0.0001 |
| **Hyperparameters for sparse compression (*compressed sparse*) (Zhu & Gupta, 2017)** | | | |
| Number of pruning iterations ($T$) | 10 | 20 | 20 |
| Number of training epochs between pruning iterations | 2 | 2 | 2 |
| Number of training epochs post-pruning | 20 | 10 | 10 |
| Number of epochs during pruning | 40 | 50 | 50 |
| Learning rate schedule during pruning (epoch range: learning rate) | 1 - 20: 0.0200
21 - 30: 0.0040
31 - 40: 0.0008 | 1 - 25: 0.0200
25 - 35: 0.0040
36 - 50: 0.0008 | 1 - 25: 0.0100
26 - 35: 0.0010
36 - 50: 0.0001 |
| **Hyperparameters for dynamic sparse reparameterization (*dynamic sparse*) (ours)** | | | |
| Number of parameters to prune ($K$) | 600 | 20,000 | 200,000 |
| Fractional tolerance of $K$ ($\delta$) | 0.1 | 0.1 | 0.1 |
| Initial pruning threshold ($H^{(0)}$) | 0.001 | 0.001 | 0.001 |
| Reparameterization period ($P$) schedule (epoch range: $P$) | 1 - 25: 100
26 - 50: 200
51 - 75: 400
76 - 100: 800 | 1 - 25: 100
26 - 80: 200
81 - 140: 400
141 - 200: 800 | 1 - 25: 1000
26 - 50: 2000
51 - 75: 4000
76 - 100: 8000 |
| **Hyperparameters for Sparse Evolutionary Training (*SET*) (Mocanu et al., 2018)** | | | |
| Number of parameters to prune at each re-parameterization step | - | 20,000 | 200,000 |
| Reparameterization period ($P$) schedule (epoch range: $P$) | - | 1 - 25: 100
26 - 80: 200
81 - 140: 400
141 - 200: 800 | 1 - 25: 1000
26 - 50: 2000
51 - 75: 4000
76 - 100: 8000 |
| **Hyperparameters for Deep Rewiring (*DeepR*) (Bellec et al., 2017)** | | | |
| $L^1$ regularization multiplier ($\alpha$) | - | $10^{-5}$ | $10^{-5}$ |
| Temperature ($T$) schedule (epoch range: $T$) | - | 1 - 25: $10^{-5}$
26 - 80: $10^{-8}$
81 - 140: $10^{-12}$
141 - 200: $10^{-15}$ | 1 - 25: $10^{-5}$
26 - 50: $10^{-8}$
51 - 75: $10^{-12}$
76 - 100: $10^{-15}$ |

## APPENDIX B    COMPARISON TO STATIC DENSE REPARAMETERIZATION METHODS

We also compared our dynamic sparse reparameterization method to a number of static dense reparameterization techniques, e.g. Denil et al. (2013); Yang et al. (2014); Moczulski et al. (2015); Sindhwani et al. (2015); Chen et al. (2015); Treister et al. (2018). Instead of sparsification, these methods impose structure on large parameter tensors by parameter sharing. Most of these methods have not been used for convolutional layers except for recent ones (Chen et al., 2015; Treister et al., 2018). We found that *HashedNet* (Chen et al., 2015) had the best performance over other static dense reparameterization methods, and also benchmarked our method against it. Instead of reparameterizing a parameter tensor with $N$ entries to a sparse one with $M < N$ non-zero components, *HashedNet*'s reparameterization is to put $M$ free parameters into $N$ positions in the parameter through a random mapping from $\{1, \cdots, N\}$ to $\{1, \cdots, M\}$ computed by cheap hashing, resulting in a dense parameter tensor with shared components.

Results of LeNet-300-100-10 on MNIST are presented in Figure 4a, those of WRN-28-2 on CIFAR10 in Figure 4b, and those of Resnet-50 on Imagenet in Table 5. For a certain global sparsity $s$ of our method, we compare it against a *HashedNet* with all reparameterized tensor hashed such that each had a fraction $1 - s$ of effective parameter count. We found that our method *dynamic sparse* significantly outperformed *HashedNet*.

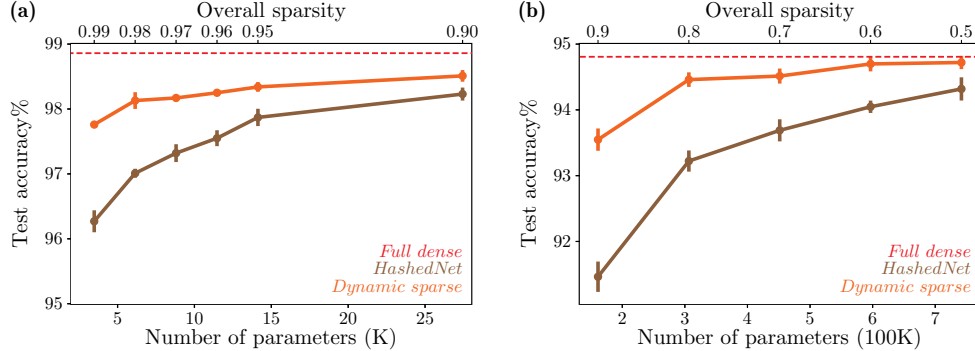

**Figure 4:** Comparison to *HashedNet*. (a) Test accuracy for LeNet-300-100-10 trained on MNIST. (b) Test accuracy for WRN-28-2 trained on CIFAR10. Conventions same as in Figure 1a.

**Table 5:** Test accuracy% (top-1, top-5) of Resnet-50 on Imagenet for *dynamic sparse* vs. *HashedNet*

| Final overall sparsity (# Parameters) | 0.8 (7.3M) | | 0.9 (5.1M) | |
|---|---|---|---|---|
| *HashedNet* | 70.0 [-4.9] | 89.6 [-2.8] | 66.9 [-8.0] | 87.4 [-5.0] |
| *Dynamic sparse* (ours) | **73.3 [-1.6]** | **92.4 [ 0.0]** | **71.6 [-3.3]** | **90.5 [-1.9]** |

Numbers in square brackets are differences from the *full dense* baseline.

APPENDIX C   A TAXONOMY OF TRAINING METHODS THAT YIELD "SPARSE" DEEP CNNS

As an extension to Section 2 of the main text, here we elaborate on existing methods related to ours, how they compare with and contrast to each other, and what features, apart from effectiveness, distinguished our approach from all previous ones. We confine the scope of comparison to training methods that produce smaller versions (i.e. ones with fewer parameters) of a given modern (i.e. post-AlexNet) deep convolutional neural network model. We list representative methods in Table 6.

We classify these methods by three key features.

**Strict parameter budget throughout training and inference**  This feature was discussed in depth in the main text. Most of the methods to date are *compression* techniques, i.e. they start training with a fully parameterized, dense model, and then reduce parameter counts. To the best of our knowledge, only three methods, namely DeepR (Bellec et al., 2017), SET (Mocanu et al., 2018) and ours, *strictly* impose, throughout the entire course of training, a fixed small parameter budget, one that is equal to

**Table 6:** Representative examples of training methods that yield "sparse" deep CNNs

| Method | Strict parameter budget throughout training and inference | Granularity of sparsity | Automatic layer sparsity |
|---|---|---|---|
| Dynamic Sparse Reparameterization (Ours) | yes | non-structured | yes |
| Sparse Evolutionary Training (SET) (Mocanu et al., 2018) | yes | non-structured | no |
| Deep Rewiring (DeepR) (Bellec et al., 2017) | yes | non-structured | no |
| NN Synthesis Tool (NeST) (Dai et al., 2017; 2018) | no | non-structured | yes |
| `tf.contrib.model_pruning` (Zhu & Gupta, 2017) | no | non-structured | no |
| RNN Pruning (Narang et al., 2017) | no | non-structured | no |
| Deep Compression (Han et al., 2015) | no | non-structured | no |
| Group-wise Brain Damage (Lebedev & Lempitsky, 2015) | no | channel | no |
| $L^1$-norm Channel Pruning (Li et al., 2016) | no | channel | no |
| Structured Sparsity Learning (SSL) (Wen et al., 2016) | no | channel/kernel/layer | yes |
| ThiNet (Luo et al., 2017) | no | channel | no |
| LASSO-regression Channel Pruning (He et al., 2017) | no | channel | no |
| Network Slimming (Liu et al., 2017) | no | channel | yes |
| Sparse Structure Selection (SSS) (Huang & Wang, 2017) | no | layer | yes |
| Principal Filter Analysis (PFA) (Suau et al., 2018) | no | channel | yes/no |

We provide examples of different categories of methods. This is not a complete list of methods.

the size of the final sparse model for inference. We make a distinction between these *direct training* methods (first block) and *compression* methods (second and third blocks of Table 6) [11].

This distinction is meaningful in two ways: (a) practically, *direct training* methods are more memory-efficient on appropriate computing substrate by requiring parameter storage of no more than the final compressed model size; (b) theoretically, these methods, if performing on par with or better than *compression* methods (as this work suggests), shed light on an important question: whether gross overparameterization is necessary for good generalization performance?

**Granularity of sparsity** The *granularity* of sparsity refers to the additional structure imposed on the placement of the non-zero entries of a sparsified parameter tensor. The finest-grained case, namely *non-structured*, allows each individual weight in a parameter tensor to be zero or non-zero independently. Early compression techniques, e.g. Han et al. (2015), and more recent pruning-based compression methods based thereon, e.g. Zhu & Gupta (2017), are non-structured (second block of Table 6). So are all direct training methods like ours (first block of Table 6).

Non-structured sparsity makes its computations difficult to accelerate on GPUs. To tackle this problem, a class of compression methods, called *structured pruning* (third block in Table 6), constrain "sparsity" to a much coarser granularity. Typically, pruning is performed to an entire convolution channel, e.g. ThiNet (Luo et al., 2017), even whole layers or residual blocks (Huang & Wang, 2017). This way, the compressed "sparse" model has essentially smaller and/or fewer *dense* parameter tensors, and computation can thus be accelerated on GPUs the same way as dense neural networks.

These *structured compression* methods, however, did not make a useful baseline in this work, for the following reasons. First, because they produce dense models, their relevance to our method (non-structured, non-compression) is far more remote than non-structured compression techniques yielding sparse models, for a meaningful comparison. Second, typical structured pruning methods substantially underperformed non-structured ones (see Table 2 for two examples, ThiNet and SSS), and emerging evidence has called into question the fundamental value of structured pruning: Mittal et al. (2018) found that the channel pruning criteria used in a number of state-of-the-art structured pruning methods performed no better than random channel elimination, and Liu et al. (2018) found that fine-tuning in a number of state-of-the-art pruning methods fared no better than direct training of a randomly initialized pruned model which, in the case of channel/layer pruning, is simply a less wide and/or less deep dense model (see Table 2 for comparison of ThiNet and SSS against *thin dense*).

In addition, we also performed extra experiments in which we constrained our method to perform "structured sparsification" and obtained significantly worse results, see Appendix D.

**Predefined versus automatically discovered sparsity levels across layers** The last key feature (rightmost column of Table 6) for our classification of methods is whether the sparsity levels of different layers of the network is automatically discovered during training or predefined by manual configuration. The value of automatic sparsification, e.g. ours, is twofold. First, it is conceptually more general because parameter reallocation heuristics can be applied to diverse model architectures, whereas layer-specific configuration has to be cognizant of network architecture, and at times also of the task to learn. Second, it is practically more scalable because it obviates manual configurations of layer-wise sparsity, agnostic of the depth and size of the network, keeping the overhead of hyperparameter tuning constant rather than scaling with model depth/size. In addition to efficiency, we also show in Appendix E extra experiments on how automatic parameter reallocation across layers contributed to its effectiveness.

In conclusion, to the best of our knowledge, our method is unique from all existing ones because it
- strictly maintained a fixed parameter footprint throughout the entire course of training, and
- automatically discovered layer-wise sparsity levels during training.

---

[11] Note that an intermediate case is NeST (Dai et al., 2017; 2018), which started training with a small network, grew it to a large size, and finally pruned it down again. Thus, a fixed parameter footprint is not strictly imposed throughout training, so we list NeST in the second block of Table 6.

## APPENDIX D  STRUCTURED VERSUS NON-STRUCTURED SPARSITY

As discussed in Appendix C, we benchmarked our direct sparse training methods against the best performing *non-structured* pruning method (Zhu & Gupta, 2017) known to us. Compression by *structured* channel-wise pruning, on the other hand, is not better than even the *thin dense* baseline (Table 2), consistent with recent criticisms (Mittal et al., 2018; Liu et al., 2018).

We asked how our method would perform if it were constrained to training sparse models at a coarser granularity. To answer this question, we performed additional experiments for WRN-28-2 trained on CIFAR10 and Resnet-50 trained on Imagenet, as described in the following.

Consider a weight tensor of a convolution layer, of size $C_{\text{out}} \times C_{\text{in}} \times 3 \times 3$, where $C_{\text{out}}$ and $C_{\text{in}}$ are the number of output and input channels, respectively. Our method performed dynamic sparse reparameterization by pruning and reallocating individual weights of the 4-dimensional parameter tensor–the finest granularity. To adapt our procedure to coarse-grain sparsity on groups of parameters, we modified our Algorithm 1 in the following ways:

1. the pruning step now removed entire groups of weights by comparing their $L^1$-norms with the adaptive threshold;
2. the adaptive threshold was updated based on the difference between the target number and the actual number of groups to prune/grow at each step;
3. the growth step reallocated groups of weights within and across parameter tensors using the same heuristic (Equation 2).

We show results at kernel-level granularity (i.e. groups are $3 \times 3$ kernels) in Figure 5 and Table 7, for WRN-28-2 on CIFAR10 and Resnet-50 on Imagenet, respectively. We made two observations. First, enforcing kernel-level sparsity leads to significantly worse accuracy compared to unstructured sparsity. Second, at this coarser granularity, our method still outperformed the *thin dense* baseline model with the same number of parameters.

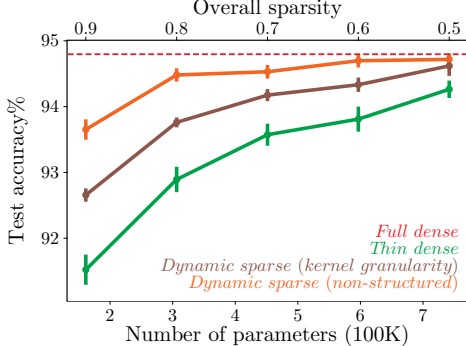

**Figure 5:** Test accuracy for WRN-28-2 trained on CIFAR10 for two variants of *dynamic sparse*, i.e. kernel-level granularity of sparsity and non-structured (same as *dynamic sparse* in the main text), as well as the *thin dense* baseline. Conventions same as in Figure 1a.

**Table 7:** Test accuracy% (top-1, top-5) of Resnet-50 on Imagenet for different levels of granularity of sparsity

| Final overall sparsity (# Parameters) | 0.8 (7.3M) | | 0.9 (5.1M) | |
|---|---|---|---|---|
| *Thin dense* | 71.6 [-3.3] | 90.3 [-2.1] | 69.4 [-5.5] | 89.2 [-3.2] |
| *Dynamic sparse (kernel granularity)* | 72.6 [-2.3] | 91.0 [-1.4] | 70.2 [-4.7] | 89.8 [-2.6] |
| *Dynamic sparse (non-structured)* | **73.3 [-1.6]** | **92.4 [ 0.0]** | **71.6 [-3.3]** | **90.5 [-1.9]** |

Numbers in square brackets are differences from the *full dense* baseline.

When we further coarsened the granularity of sparsity to channel level (i.e. groups are $C_{\text{in}} \times 3 \times 3$ slices that generate output feature maps), our method utterly failed to produce performant models.

## APPENDIX E    AUTOMATIC PARAMETER REALLOCATION ACROSS LAYERS ENABLES LEARNING AT EXTREME SPARSITY LEVELS

In order to assess whether and how much our parameter reallocation heuristic (Equation 2) contributed to the effectiveness of our method, we did a set of control experiments in which all free parameter reallocation were constrained to within parameter tensors, i.e. parameter reallocation across layers were disabled, and sparsity stayed constant (uniformly initialized) for each layer.

The results are presented in Figure 6 for LeNet-300-100-10 on MNIST. We found that removing inter-layer parameter allocation yielded worse performance, particularly at extremely high sparsity levels. This suggested that parameter reallocation across the entire model was beneficial to our dynamic reparameterization.

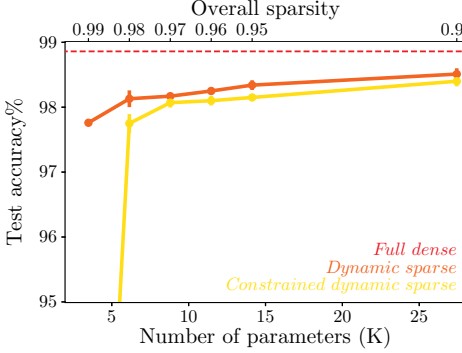

**Figure 6:** Test accuracy for LeNet-300-100-10 on MNIST of *dynamic sparse* (i.e. Algorithm 1) compared against *constrained dynamic sparse* for which parameter reallocation occurred only within, but not across, layers.

