# OpenReview forum: "Parameter efficient training of deep convolutional neural networks by dynamic sparse reparameterization"
_ICLR.cc/2019/Conference_

### Official Review · AnonReviewer1 · 2018-11-02
**Efficient dynamic reparameterization but the claims need to be revisited**

**Rating:** 6
**Confidence:** 4

**Review:**

The paper provides a dynamic sparse reparameterization method allowing small networks to be trained at a comparable accuracy as pruned network with (initially) large parameter spaces. Improper initialization along with a fewer number of parameters requires a large parameter model, to begin with (Frankle and Carbin, 2018).  The proposed method which is basically a global pooling followed by a tensorwise growth allocates free parameters using an efficient weight re-distribution scheme, uses an approximate thresholding method and provides automatic parameter re-allocation to achieve its goals efficiently. The authors empirically demonstrate their results on MNIST, CIFAR-10, and Imagenet and show that dynamic sparse provides higher accuracy than compressed sparse (and other) networks.

The paper is addressing an important problem where instead of training and pruning, directly training smaller networks is considered. In that respect, the paper does provide some useful tricks to reparameterize and pick the top filters to prune. I especially enjoyed reading the discussion section.

However, the hyperbole in claims such as "first dynamic reparameterization method for training convolutional network" makes it hard to judge the paper favorably given previous methods that have already proposed dynamic reparameterization and explored. This language is consistent throughout the paper and the paper needs a revision that positions this paper appropriately before it is accepted.

The proposed technique provides limited but useful contributions over existing work as in SET and DeepR. However, an empirical comparison against them in your evaluation section can make the paper stronger especially if you claim your methods are superior.

How does your training times compare with the other methods? Re-training times are a big drawback of pruning methods and showing those numbers will be useful.

---

> ### Author Response · Authors · 2018-11-24
> **Response to Reviewer**
>
>
> Thank you for the review.   We have substantially revised the manuscript with the following major changes:
>
> 1. Contributions in Introduction (also the rest of the manuscript) completely rewritten to state novelty accurately
> 2. Inclusion of results of additional performance benchmark against existing methods, DeepR and SET
> 3. Inclusion of results of computational cost benchmarked against existing methods, DeepR and SET
> 4. Revised Experimental Results section and two additional appendices that further expanded the scope of comparison to structured compression methods
>
> We hope the improved manuscript is worthy of publication now.
>
> Our response to your specific comments:
>
> 1. We have rewritten the entire manuscript to state our novelty accurately.  We now make only two claims in contributions, which are carefully limited to the exact scope of this investigation.  We specifically eliminated claims of "first".
>
> 2. We have strengthened the manuscript by including results of full comparisons to SET and DeepR (the new Figure 2, Table 2) to support our main claim that ours outperformed these methods.
>
> 3. We have included a new Table 3, containing the computational overhead of our parameter reallocation in comparison to those of DeepR and SET.  Indeed, re-training times are a drawback of pruning methods. We now state clearly that our method actually runs for a fewer number of epochs than pruning-based methods--our method only trained for the same number of epochs as the original training of the large dense model in pruning methods minus the additional retraining (see the revised Appendix A).
>
> * Note: Due to the high computational requirements of DeepR, the results for DeepR on resnet50 were not available in time for the revision submission. We include the results (top-1, top-5 accuracy) below. The accuracy of DeepR lags behind our method and behind SET.
>
> +----------------------------------------------------------------------------------------------+
> | Method                                    |     Sparsity = 0.9     |     Sparsity = 0.8     |
> +----------------------------------------------------------------------------------------------+
> | Mocanu et al. 2018 (SET)      |         70.4, 90.1        |         72.6, 91.2        |
> +----------------------------------------------------------------------------------------------+
> | Bellec et al. 2017 (DeepR)      |         70.2, 90.0        |         71.7, 90.6        |
> +----------------------------------------------------------------------------------------------+
> | Ours                                         |         71.6, 90.5        |         73.3, 92.4        |
> +----------------------------------------------------------------------------------------------+

---

> > ### Comment · AnonReviewer1 · 2018-11-28
> > **Thanks for the update!**
> >
> > Thanks for the comprehensive revision as well as providing new experiments and comparisons. Your revision adequately addresses the concerns I raised in my original review.

---

### Official Review · AnonReviewer2 · 2018-11-03
**In its present form, this paper seems more like engineered modifications of existing pipelines with insufficient validation, rather than a mature research contribution.**

**Rating:** 4
**Confidence:** 4

**Review:**

This paper presents a method for training neural networks where an efficient sparse/compressed representation is enforced throughout the training process, as opposed to starting with are large model and pruning down to a smaller size.  For this purpose a dynamic sparse reparameterization heuristic is proposed and validated using data from MNIST, CIFAR-10, and ImageNet.

My concerns with this work in its present form are two-fold.  First, from a novelty standpoint, the proposed pipeline can largely be viewed as introducing a couple heuristic modifications to the SET procedure from reference (Mocanu, et al., 2018), e.g., substituting an approximate threshold instead of sorting for removing weights, changing how new weights are redistributed, etc.  The considerable similarity was pointed out by anonymous commenters and I believe somewhat understated by the submission.  Regardless, even if practically effective, these changes seem more like reasonable engineering decisions to improve the speed/performance rather than research contributions that provide any real insights.  Moreover, there is no attendant analysis regarding convergence and/or stability of what is otherwise a sequence of iterates untethered to a specific energy function being minimized.

Of course all of this could potentially be overcome with a compelling series of experiments demonstrating the unequivocal utility of the proposed modifications.  But it is here that unfortunately the paper falls well short.  Despite its close kinship with SET, there are surprisingly no comparisons presented whatsoever.  Likewise only a single footnote mentions comparative results with DeepR (Bellec et al., 2017), which represents another related dynamic reparameterization method.  In a follow up response to anonymous public comments, some new tests using CIFAR-10 data are presented, but to me, proper evaluation requires full experimental details/settings and another round of review.

Moreover, the improvement over SET in these new results, e.g., from a 93.42 to 93.68 accuracy rate at 0.9 sparsity level, seems quite modest.  Note that the proposed pipeline has a wide range of tuning hyperparameters (occupying a nearly page-sized Table 3 in the Appendix), and depending on these settings relative to SET, one could easily envision this sort of minor difference evaporating completely.  But again, this is why I strongly believe that another round of review with detailed comparisons to SET and DeepR is needed.

Beyond this, the paper repeatedly mentions significant improvement over "start-of-the-art sparse compression methods." But this claim is completely unsupported, because all the tables and figures only report results from a single existing compression baseline, namely, the pruning method from (Zhu and Gupta, 2017) which is ultimately based on (Han et al., 2015).  But just in the last year alone there have been countless compression papers published in the top ML and CV conferences, and it is by no means established that the pruning heuristic from (Zhu and Gupta, 2017) is state-of-the-art.

Note also that reported results can be quite deceiving on the surface, because unless the network structure, data augmentation, and other experimental design details are exactly the same, specific numbers cannot be directly transferred across papers.  Additionally, numerous published results involve pruning at the activation level rather than specific weights.  This definitively sacrifices the overall compression rate/model size to achieve structured pruning that is more naturally advantageous to implementation in practical hardware (e.g., reducing FLOPs, run-time memory, etc.).  One quick example is Luo et al., "ThiNet: A Filter Level Pruning Method for Deep Neural Network Compression," ICCV 2017, but there are many many others.

And as a final critique of the empirical section, why not report the full computational cost of training the proposed model relative to others?  For an engineered algorithmic proposal emphasizing training efficiency, this seems like an essential component.


In aggregate then, my feeling is that while the proposed pipeline may eventually prove to be practically useful, presently this paper does not contain a sufficient aggregation of novel research contribution and empirical validation.

Other comments:

- In Table 2, what is the baseline accuracy with no pruning?

- Can this method be easily extended to prune entire filters/activations?

---

> ### Author Response · Authors · 2018-11-24
> **Response to Reviewer (Part 2/2)**
>
>
> 4. On benchmark against post-training compression baselines:  Thank you for raising this important point.  We agree that the line of work from Han et al. 2015 to Zhu & Gupta 2017 is only a subset of all compression methods.  Even though Zhu & Gupta 2017 is the strongest sparse compression baseline known to us, we now state clearly that we close the performance gap to the iterative pruning method of Zhu and Gupta 2017, instead of saying "compression methods" in general.
>
> 5. On structured compression method such as ThiNet:  In the previous version of the manuscript, we did not benchmark against structured compression method such as ThiNet because they (a) produce dense instead of sparse models, and (b) significantly underperformed non-structured compression, such as Zhu & Gupta 2017, despite their efficiency on GPUs.  In the revision, we made the following changes to address this issue: (a) we included comparisons against two representative structured pruning methods in Table 2; (b) we included a new Appendix C to compare and contrast a wide range of methods, painting a broad picture of relevant existing methods to show where our method stands; (c) we did additional experiments to impose group structure on sparsity using our method, and show degraded results (the new Appendix D); (d) we specifically discussed the issue of structured versus non-structured sparsification, and its implications for optimal computing hardware architecture (last two paragraphs of the Discussion section).
>
> 6. On the difficulty of comparing results across papers:  In the revision we included our own experiments of DeepR and SET, carefully controlled for comparison to ours.  For comparison with ThiNet (Luo et al. 2017) and SSS (Huang & Wang 2017), we adapted the results from the original papers (See the new Table 2).  To ameliorate the potential minor differences in experimental protocols, we also report the relative difference from the full dense model performance reported in that same paper (square brackets in the new Table 2)--comparison of methods can now be based on how much accuracy degradation from a controlled baseline a method introduces, rather than on absolute accuracy figures.
>
> 7. On computational cost:  We now include quantifications of computational overhead of our method, DeepR and SET, in the last paragraph of the Experimental Results section and in Table 3.
>
> 8. On other comments:  (a) We included the full dense baseline in the new Table 2 (rightmost column).  (b) We included a new Appendix D to present extra experiments where we applied our methods to group pruning of 3x3 kernels.  We show that this led to a significant but minor degradation in performance.  We also discussed the pros and cons of structured vs. non-structured sparsification in Discussion and Appendix C as stated above.
>
> * Note: The current PDF of the manuscript has blanked DeepR entry in Table 3.  Due to the high computational requirements of this experiment, it is still running.  We will fill in the numbers as soon as they are available.

---

> > ### Comment · AnonReviewer2 · 2018-11-29
> > **comments to the author rebuttal**
> >
> > Although the experimental section has been expanded a bit and the overall paper has been upgraded, I still believe that the novelty of this work is limited.  As mentioned in my original review, the proposed pipeline represents reasonably engineered modifications of the existing SET pipeline, but there are no significant insights beyond this.  Moreover, the added heuristics often seem to be marginally important.  For example, if we look at Figure 2(a) in the revision, the proposed algorithm has almost the same performance as SET. For these and other reasons below, I continue to believe then that this work is perhaps below the bar for ICLR, while acknowledging the effort it takes to engineer and test this type of model.
> >
> > Other lingering concerns with this work:
> >
> > - With regard the hyperparameter tuning, the rebuttal comments state that "we did not tune them and simply used the same hyperparameter settings in the original papers where these models were described."  This seems to reinforce the notion that this is not a fundamentally different model, but one intimately related to existing papers.
> >
> > - The revised version claims in the introduction that the proposed method is more efficient than existing dynamic sparse reparameterization training techniques.  But this seems like an overstatement, because if we look at Table 3, the proposed method requires almost exactly the same computational complexity as the existing SET method (the difference is only in the forth significant digit).
> >
> > - The rebuttal states that "(Zhu & Gupta, 2017) is the strongest sparse compression baseline known to us."  But as I mentioned previously, there have been numerous DNN compression methods introduced at multiple major ML and CV conference over the last year, and it is essential to check these proceedings to become aware of the latest developments.  This is especially true for an empirical paper of this type without any analytical contribution.  Also, the revision continues to imply without justification that (Zhu & Gupta, 2017) represents the state-of-the-art (see bullet point 4 on page 5).
> >
> > - According to the rebuttal, some DeepR experiments are still running and therefore could not be included in the revision.  But this is not an excusable omission, because as an obvious, direct competitor to the proposed method, these results should have been present in the original submission.
> >
> > - As mentioned in my original review, the proposed algorithmic steps are not minimizing any particular energy function per se, and yet there is no discussion of convergence or stability.  This was not addressed in the rebuttal.

---

> > > ### Author Response · Authors · 2018-11-30
> > > **Authors' response (Part 2 of 2)**
> > >
> > >
> > > - With regard to "numerous DNN compression methods introduced at multiple major ML and CV conference over the last year", we did a comprehensive survey of compression papers, which are listed in Appendix C, together with their properties.  We still see that the best compression performance is achieved by Zhu and Gupta, 2017.  With adequate due diligence we honestly claim that `tensorflow.contrib.model_pruning` (Zhu and Gupta, 2017) is a "state-of-the-art" sparse compression method "known to us" at the time of this paper being written.  We would appreciate it if you could provide us with a specific stronger baseline, demonstrably stronger than Zhu & Gupta 2017, and we are happy to benchmark our method against it and call it "state-of-the-art".  Absent such a method in the literature that you could refer to us, the claim that we did not compare ours to the best sparse compression method in existence is a rather weak one, and unfair.  Note that most recent pruning techniques deals with structured or filter-wise sparsity that does not directly compare with our method and significantly underperformed non-structured pruning (see Appendix C, D, see Liu et al. 2018).
> > >
> > > - We believe there is a misunderstanding regarding your response on hyperparameter tuning in writing "This seems to reinforce the notion that this is not a fundamentally different model, but one intimately related to existing papers."  We referred to "hyperparameters for training", such as learning rate, momentum, L1/L2 decay, etc..., these were taken the same as those presented in the original paper (e.g. He et al. 2015) where the model (e.g. Resnet-50) was presented.  These do not include hyperparameters for sparse reparameterization (which is the subject of this paper and we did tune those).  We do not believe this is related to the novelty of our method at all--our paper does not produce "fundamentally different model"s, but trains sparse versions of existing, successful models, such as Resnet.  If anything, using the exact hyperparameters for training as in the original papers only strengthens our method--our method is robust enough that we do not need to re-tune hyperparamters for training it and it just worked well with the original network's hyperparameters.
> > >
> > > - With regard to DeepR, we did demonstrate a significant accuracy/performance advantage over DeepR for the case of WRN-28-2 on CIFAR10.  Experiment on Imagenet is much more challenging due to the high computational cost of DeepR, but we agree that it should be (and will be) part of the paper (note that the DeepR paper never attempted experiments at this scale).  With concrete evidence, our results on CIFAR10, together with the significantly faster training times of our method are strong indicators that our method is superior to DeepR in both accuracy and speed.  We agree with you that Table 3 shows ours and SET both introduce negligible computational overhead, but our method has other advantages over SET (e.g. producing better accuracy, able to train at high sparsity levels), besides the slight reduction in computational cost.

---

> > > > ### Author Response · Authors · 2018-12-03
> > > > **Authors' response (DeepR results on resnet50)**
> > > >
> > > > Due to the high computational requirements of DeepR, the results for DeepR on resnet50 were not available in time for the revision submission. We include the results (top-1, top-5 accuracy) below. The accuracy of DeepR lags behind our method and behind SET.
> > > >
> > > > +----------------------------------------------------------------------------------------------+
> > > > | Method                                    |     Sparsity = 0.9     |     Sparsity = 0.8     |
> > > > +----------------------------------------------------------------------------------------------+
> > > > | Mocanu et al. 2018 (SET)      |         70.4, 90.1        |         72.6, 91.2        |
> > > > +----------------------------------------------------------------------------------------------+
> > > > | Bellec et al. 2017 (DeepR)      |         70.2, 90.0        |         71.7, 90.6        |
> > > > +----------------------------------------------------------------------------------------------+
> > > > | Ours                                         |         71.6, 90.5        |         73.3, 92.4        |
> > > > +----------------------------------------------------------------------------------------------+

---

> > > > > ### Author Response · Authors · 2018-12-12
> > > > > **Would you please respond to our rebuttal?**
> > > > >
> > > > >
> > > > > Thank you again for your useful comments, which were common concerns raised by all 3 reviewers, including (1) lack of comparison to prior work, and (2) inaccurate claims of contributions.
> > > > >
> > > > > In the revision submitted together with a point-by-point rebuttal to your review on Nov. 23, we have fully addressed these concerns, and Reviewer #3 found our revision satisfactory.
> > > > >
> > > > > As it has been weeks since our previous response, we wonder whether you would share the same view on the revised manuscript's suitability for publication, or if there were still lingering concerns, we would appreciate it if you could reply to our most recent point-by-point rebuttal with specifics so that we could address them.
> > > > >
> > > > > Thank you very much!

---

> > > ### Author Response · Authors · 2018-11-30
> > > **Authors' response (Part 1 of 2)**
> > >
> > >
> > > Thank you for your comments.  We believe the argument on the lack of novelty lacks factual support.  Our point-by-point response:
> > >
> > > - "Moreover, the added heuristics often seem to be marginally important... Figure 2(a) ...almost the same performance as SET" This argument disregards key evidence for the opposite.  First, our algorithm did lead to significantly better accuracy than SET on all the benchmarks we tested, no matter how small the differences are in certain cases.  The differences are in fact more substantial in other cases.  For instance, the improvement on Imagenet is more prominent (Table 2).  Second, the improvement is especially more apparent at high sparsity levels where SET failed catastrophically while our method does not (see Figure 6 in appendix E).  Finally, our automatic parameter allocation heuristic that discovered the number of parameters to allocate to each layer is entirely novel.
> > >
> > > - Even if our method lacked any novelty or performance improvement as compared to SET (which it does not as our revised manuscript shows with quantitative evidence), it would still be a significant stride forward from Mocanu et al. 2018 just to apply their exact same method to show its usefulness in training deep sparse convolution nets such as Resnet-50 in action on large datasets like Imagenet.  In other words, Mocanu et al. 2018 did not even show if SET is at all applicable to convolution layers, let alone deep Resnets.
> > >
> > > - We agree with you that a theoretical guarantee of convergence and stability is not presented, and the heuristic could not be cast into an optimization procedure of an objective function.  But we do not think this is an adequate reason to dismiss an empirical paper.  Convergence guarantees are hard to come by in deep networks where there are no convergence guarantees of even basic stochastic gradient descent in most real-world cases.  We practiced empirical rigor in the paper: we exhaustively described our method and experiments in detail, empirically validated our method's performance and stability, and publicized all source code for all experiments, a standard of which even the papers you cited in your review (e.g. Mocanu et al. 2018) fell shy (note that there is no theoretical guarantee of SET's stability in that paper either).  Rigorous empirical findings is beneficial to the field because the dissemination of knowledge on "what works" would eventually lead to theory on "why it works".

---

> ### Author Response · Authors · 2018-11-24
> **Response to Reviewer (Part 1/2)**
>
>
> Thank you for the review.   We have substantially revised the manuscript with the following major changes:
>
> 1. Contributions in Introduction (also the rest of the manuscript) completely rewritten to state novelty accurately
> 2. Inclusion of results of additional performance benchmark against existing methods, DeepR and SET
> 3. Inclusion of results of computational cost benchmarked against existing methods, DeepR and SET
> 4. Revised Experimental Results section and two additional appendices that further expanded the scope of comparison to structured compression methods
>
> We hope the improved manuscript is worthy of publication now.
>
> Our response to your specific comments:
>
> 1. On benchmarking against existing methods:  We fully agree with you on the weakness of the manuscript due to lack of quantitative comparisons to prior work.  We believe this revision adequately rectified it.  Specifically, we performed additional WRN-28-2 on CIFAR10 and Resnet-50 on Imagenet experiments for DeepR and SET.  The results are presented in the new Figure 2, Table 2 and Table 3.  We now show with concrete evidence that our method outperformed both DeepR and SET.
>
> 2. On hyperparameter tuning:  For hyperparameters of training the models, we did not tune them and simply used the same hyperparameter settings in the original papers where these models were described.  For hyperparameters of reparameterization by DeepR and SET, because the original papers did not attempt experiments at the same scale, we did a hyperparameter sweep for DeepR and reported the best result; for SET, in order to make a fair comparison, we used the exact same hyperparameters for both SET and ours.  These facts were not clearly stated in the previous version, but now clearly stated together with the list of hyperparameters (Table 4) in the revision.
>
> 3. On our method's similarity to SET:  Our method is inspired by SET, based on the similar sparse reparameterization mechanism, but with an adaptive threshold and a heuristic for automatic parameter reallocation across layers.  These differences might seem incremental, but we believe our work made a substantial stride forward from what was reported in Mocanu et al. 2018, for the following reasons (as now stated in the revised manuscript). (a) Our method did produce significantly better generalizing sparse models than SET, and fully closed the performance gap toward compression by iterative pruning, of which SET in some cases still fell short (we now provide quantitative results).  (b) Automatic parameter reallocation without manual configuration of sparsity per layer makes sparse training much more scalable: the burden of hyperparameter tuning is constant instead of scales with network depth.  (c) Finally, Mocanu et al. 2018 demonstrated their method on multi-layer perceptrons on MNIST.  We believe scaling up to deep convolutional networks such as Resnet-50 and to large datasets such as Imagenet is not just a trivial increment.

---

### Official Review · AnonReviewer3 · 2018-11-05
**The authors designed a dynamic reparameterization method to apply model compression in deep neural architectures. They compared their proposed framework with three baseline methods in terms of test accuracy and sparsity.  The comparison to the existing works is lacking.**

**Rating:** 4
**Confidence:** 4

**Review:**

Weaknesses:

1-The authors claim that: "Compared to other dynamic reparameterization methods that reallocate non-zero parameters during training, our approach broke free from a few key limitations and achieved much better performance at lower computational cost." => However, there is no quantitative experiments related to other dynamic reparameterization methods. There should be at least sparsity-accuracy comparison to claim achieving better performance. I expect authors compare their work at-least with with DEEP R, and NeST even if it is clear for them that they produce better results.
2-The second and fourth contributions are inconsistent: In the second one, authors claimed that they are the first who designed the dynamic reparameterization method. In the fourth contribution, they claimed they outperformed existing dynamic sparse reparameterization.  Moreover, it seems DEEP R also is a  dynamic reparameterization method because DEEP R authors claimed: "DEEP R automatically rewires the network during supervised training so that connections are there where they are most needed for the task, while its total number is all the time strictly bounded."
3- The authors claimed their proposed method has much lower computational costs, however, there is no running time or scalability comparison.


Suggestions:
1-Authors need to motivate the applications of their work. For instance, are they able to run their proposed method on mobile devices?
2-For Figure 2 (c,d) you need to specify what each color is.
3-In general, if you claim that your method is more accurate or more scalable you need to provide quantitative experiments. Claiming is not enough.
4-It is better to define all parameters definition before you jump into the proposed section. Otherwise, it makes paper hard to follow.  For instance, you didn't define the R_l directly (It is just in the Algorithm 1).

---

> ### Author Response · Authors · 2018-11-24
> **Response to Reviewer**
>
>
> Thank you for the review.  We have substantially revised the manuscript with the following major changes:
>
> 1. Contributions in Introduction (also the rest of the manuscript) completely rewritten to state novelty accurately
> 2. Inclusion of results of additional performance benchmark against existing methods, DeepR and SET
> 3. Inclusion of results of computational cost benchmarked against existing methods, DeepR and SET
> 4. Revised Experimental Results section and two additional appendices that further expanded the scope of comparison to structured compression methods
>
> We hope the improved manuscript is worthy of publication now.
>
> Our response to your comments on weaknesses:
>
> 1. The revised manuscript now includes direct quantitative comparisons to all direct sparse training techniques with a strict parameter budget (i.e. DeepR and SET), for the deep residual net experiments (on CIFAR10 and Imagenet, see Figure 2 and Table 2).  We did not include NeST because NeST does not impose a strict parameter budget during training--it grows a small network to a large one and then prunes it down.  Our claim here is that our method yielded the best accuracy given a strictly fixed parameter budget throughout training so only DeepR and SET are relevant to this claim.  We further explain in full detail the relationships between our method and numerous others in a new Appendix C.
>
> 2. We apologize for the confusion.  The claim was indeed worded incorrectly.  The correct claim is that we are the first to apply sparse dynamic reparameterization to training of large CNNs (such as Resnets) on large datasets, because previous methods of the same kind were demonstrated only on small networks.  We have completely rewritten the contributions with this claim removed.
>
> 3. Per your suggestion, we added a last paragraph to the Experimental Results section and included a new Table 3 with numbers to support our claim on efficiency.  Our scalability argument is supported by the fact that our method discovers layer-wise sparsity automatically during training without the need to predefine sparsity per layer by manual configuration as required by other methods, so that the cost of hyperparameter tuning is constant instead of scaling with network depth.
>
> Our response to your suggestions:
>
> 1. Per your suggestion, we revised the Introduction section and included the follwing sentence: "... a dynamic sparse reparameterization technique able to train sparse models de novo without the need to compress a large model, a desirable feature for training on memory- and power-constrained devices."  Furthermore, a related, more nuanced point on hardware-efficiency was discussed in the last two paragraphs of the Discussion section.
>
> 2. We rectified the unnecessary use of color and made these panels grayscale with specific text labels.
>
> 3. We have included new results (see the new Figure 2, Table 2 and Table 3) and revised the text to provide concrete support of the claims.
>
> 4. We now defined these parameters in the text (in the revised Methods section) in addition to in Algorithm 1.
>
> * Note: Due to the high computational requirements of DeepR, the results for DeepR on resnet50 were not available in time for the revision submission. We include the results (top-1, top-5 accuracy) below. The accuracy of DeepR lags behind our method and behind SET.
>
> +----------------------------------------------------------------------------------------------+
> | Method                                    |     Sparsity = 0.9     |     Sparsity = 0.8     |
> +----------------------------------------------------------------------------------------------+
> | Mocanu et al. 2018 (SET)      |         70.4, 90.1        |         72.6, 91.2        |
> +----------------------------------------------------------------------------------------------+
> | Bellec et al. 2017 (DeepR)      |         70.2, 90.0        |         71.7, 90.6        |
> +----------------------------------------------------------------------------------------------+
> | Ours                                         |         71.6, 90.5        |         73.3, 92.4        |
> +----------------------------------------------------------------------------------------------+

---

> > ### Author Response · Authors · 2018-12-12
> > **Would you please respond to our rebuttal?**
> >
> >
> > Thank you again for your useful comments, which were common concerns raised by all 3 reviewers, including (1) lack of comparison to prior work, and (2) inaccurate claims of contributions.
> >
> > In the revision submitted together with a point-by-point rebuttal to your review on Nov. 23, we have fully addressed these concerns, and Reviewer #3 found our revision satisfactory.
> >
> > As it has been weeks since our previous response, we wonder whether you would share the same view on the revised manuscript's suitability for publication, or if there were still lingering concerns, we would appreciate it if you could reply to our most recent point-by-point rebuttal with specifics so that we could address them.
> >
> > Thank you very much!

---

### Public Comment · (anonymous) · 2018-10-15
**Issues with claimed contributions**

We disagree with some of the claims made by this paper.

Claim 1: “Ours is the first systematic method able to train sparse models directly without an increased parameter footprint during the entire course of training, and still achieve performance on par with post-training compression of dense models, the best result at a given sparsity.”

The authors of [2], which introduces DeepR, compare their technique to l1-shrinkage and magnitude-based pruning and demonstrate on-par or better performance than each for a given sparsity.

DeepR achieves the same bounded parameter footprint as the technique presented here, and does not appear to have been evaluated beyond the experiments in the original publication, yet the authors do not compare to this technique. Given this, it seems premature for this work to claim that they have achieved something that existing techniques cannot, especially considering [2] demonstrates that they achieve performance on par or better than the compression techniques they compare to.

Claim 2: “We described the first dynamic reparameterization method for training convolutional network.”

The original DeepR paper demonstrates results on a convolutional neural network. This paper cites the original DeepR paper and refers to it as a “dynamic sparse reparameterization” technique.

Claim 4: “Our method not only outperformed existing dynamic sparse reparameterization techniques, but also incurred much lower computational costs”

This work does not compare to any existing dynamic sparse reparameterization technique. They also do not measure the runtime of their technique or compare to the baseline sorting-based pruning (e.g., TensorFlow model pruning [3]).

In addition to these issues with the claimed contributions of this paper, the introduced “dynamic sparse reparameterization” technique only differs from Sparse Evolutionary Training (SET) [1] in its use of an approximate threshold for removing weights and in how it redistributes weights after pruning. Both of these modifications are potentially valuable contributions, but the authors make very broad claims rather than list these modifications and demonstrate their value over the existing methods.

References
1. https://www.nature.com/articles/s41467-018-04316-3.pdf
2. https://arxiv.org/pdf/1711.05136.pdf
3. https://github.com/tensorflow/tensorflow/tree/master/tensorflow/contrib/model_pruning

---

> ### Author Response · Authors · 2018-10-15
> **Authors response**
>
> *On Claim 1:*
> Indeed, DeepR showed that it performs better than L1-shrinkage and magnitude-based pruning. However, we use as our compression baseline the iterative pruning technique introduced in Zhu et al. 2017 that gives a stronger baseline and outperforms DeepR as shown by the accuracies in Fig.1a (compare to Fig. 3A in the DeepR paper). Our method closely matches the performance of this stronger baseline (see Fig.1a). Our claim is thus accurate given the stronger compression baseline we used.  We will rewrite the claim and make it more accurate by highlighting that it refers to the compression baseline obtained using the method in Zhu et al. 2017, instead of using the more general term "post-training compression".
>
> For the MNIST network, we do indeed compare to one of the few numbers presented in the DeepR paper (see footnote 6 on page 5) and show better performance.  We rolled our own implementation of DeepR and found that evaluating DeepR on large imagenet-class networks was very computationally expensive (5x slower than our approach). Moreover, DeepR has more hyper-parameters than our approach involving, for example, an annealing schedule for the parameter update noise, layer-wise regularization coefficients, and hand-tuned layer sparsities. The large number of hyper-parameters, together with the slowness of DeepR, make a well-tuned evaluation on large networks extremely challenging. The authors of the DeepR paper do not provide code, nor guidelines on how to use DeepR in deep all-convolutional networks. We thus limit our comparison to the MNIST case.
>
>
> *On Claim 2:*
> We apologize that the wording of this claim is not specific enough. In this claim, we were referring to modern all-convolutional networks such as residual networks. Our experiments showed that DeepR was very slow for these networks in the imagenet case. In the DeepR paper, DeepR was also not applied to the entire convolutional network, but only to a specific layer while other layers were left dense. Ours is the first application of dynamic reparameterization to an entire convolutional network. We fully acknowledge, however, the fact that DeepR was the first dynamic method of the kind applied to a convolutional layer. In the next revision, we will make the wording sufficiently clear to reflect the above facts.
>
> *On Claim 4:*
> We achieve better performance than DeepR on the small MNIST network. For the bigger convolutional networks, DeepR was very slow (see below). We will make the accuracy claim more precise by limiting it to the MNIST case (which is the case in which we can feasibly run and compare to DeepR). We make our claim regarding computational cost based on 2 observations:
> 1)DeepR runs the re-wiring step every iteration while we re-allocate/rewire every few hundreds/thousands of iterations. DeepR also needs to generate a Gaussian random number at each parameter update which incurs extra MAC operations. We will add numbers to the paper to exactly quantify the extra operation needed by DeepR (due to increased rewire frequency and the need for Gaussian random number generation for each parameter update)
> 2)We implemented DeepR ourselves. For imagenet training on 4 Titanxp GPUs, training using DeepR was 5x slower than our approach. We will include this DeepR implementation with our code release.
>
> We did compare against sorting-based pruning methods, because Tensorflow model pruning [3] was based on the sparse compression technique described by Zhu et al. 2017, which was the strongest baseline (called "compressed sparse" in the manuscript) we benchmarked against in the paper.
>
> Indeed, runtime is an important metric. We did not observe significant slow-down when using our parameter-reallocation method (since it is only applied sporadically during training). We will include the wall-time runtime of our approach compared to training without parameter-reallocation to make this observation more precise. We will also include a comment on how the runtime of our method compares to DeepR.
>
> Earlier methods like DeepR and SET provided key inspirations for this work. However, we have gone further than these previous algorithms and presented concrete results addressing several of their limitations (the computational inefficiency of DeepR, and its need for pre-specified layer sparsities; and the computational inefficiency of SET involving sorting, and the need for pre-specified layer sparsities). We illustrated for the first time the applicability of dynamic re-parameterization to practical large-scale convolutional networks, which earlier dynamic reparameterization approaches were not scalable enough to handle.  Thank you again and we are happy to address your further comments.

---

> > ### Public Comment · (anonymous) · 2018-10-19
> > **Response**
> >
> >
> > Claim 1:
> > Zhu et al. 2017 and DeepR were both submissions to ICLR 2018, thus it is not possible for DeepR to have compared to their pruning technique. They compared to a strong pruning baseline and beat it, claiming that you are the first because your performance exceeds a baseline they could not have compared to doesn’t make sense.
> >
> > If your claim is that DeepR or SET cannot achieve comparable performance to the technique of Zhu et al. you should demonstrate this experimentally and include the results in your paper. A single data point of comparison in a footnote is not sufficient to establish superiority. If you were to demonstrate that DeepR and SET cannot achieve performance that your method can, this is a very valuable contribution and you can explain this and include it as a claim.
> >
> > Also, 5x slowdown is not so absurd that it precludes comparison given that ImageNet can be trained in under a day. You also pointed out that you have already re-implemented their technique. The authors of DeepR do provide code (see the very first line of the appendix). It is not clear why you should have any issue applying it to a deep all-convolutional network, the technique is very straightforward and is agnostic to model architecture. You also provide no comparison to SET, which your technique mirrors very closely.
> >
> > Claim 2:
> > If your claim is that this is the first application of a “dynamic sparse reparameterization” to an entire convolutional network, it is not clear why this is a valuable contribution. DeepR and SET can both be trivially applied to an entire convolutional network.
> >
> > Claim 4:
> > As stated above, a 5x slowdown is not so large that it justifies not comparing to a technique when claiming superiority. And a single data point in a footnote is not a sufficient comparison.
> >
> > For your computational complexity claim, you need to also compare to SET. It is not surprising that DeepR is very slow, and SET will almost certainly be much faster than it. Your technique only differs from SET in your use of an approximate threshold and your weight redistribution scheme. You claim that your technique is faster than SET because of this, but you provide no data to back up this claim.
> >
> > When you do measure the performance of these techniques, it is important to note that the number of iterations between pruning steps is a trivial hyperparameter that can be adjusted for any pruning technique (and is commonly; see TensorFlow model pruning), and that you should compare to these techniques with the same number of iterations between pruning steps.
> >
> > Also, you claim to compare to the technique of Zhu et al., but the sparsity function you use (listed in the appendix) is not the same as theirs. If you want to compare to their technique, you should use TensorFlow model pruning, which is what was used in their experiments.
> >
> > We do not dispute the potential value of your approximate thresholding technique and weight redistribution technique. Our issues are that a) you do not properly compare to existing techniques to establish that either of these modifications has provided an improvement and b) you claim significant novelty relative to these techniques, in particular to SET, which is extremely close to your method and to which you do not compare at all.

---

> > > ### Author Response · Authors · 2018-10-24
> > > **Authors' response (1/2)**
> > >
> > >
> > > Thank you for your comments.  Please find our full response below.
> > >
> > > **Claim 1:**
> > > We do not intend to discount the claims of either Zhu et al. 2017 or Bellec et al. 2017 (DeepR), which are two concurrent papers in ICLR 2018.  Being a submission to ICLR 2019, our work had the opportunity to benchmark against both techniques.  We found that Zhu et al. 2017 happened to be a stronger baseline, and this was the reason underlying our decision of using it as a previous state-of-the-art benchmark in our paper.  In fact, we did compare our method to various previous methods, including DeepR and SET; the reason why we did not include the comparison results in the manuscript was because (1) they did not beat the stronger network compression baseline, and (2) code and hyperparameter settings for these methods for the experiments we did were not available publicly, even though we made our own implementation based on the original papers and our discussion with authors (for performance metrics available in original papers we did include side-by-side comparisons in our manuscript, e.g. performance of LeNet-300-100 at 99% sparsity on MNIST reported by the DeepR paper).  Techniques like SET that do not use parameter reallocation among layers fared worse than our technique on MNIST as shown by Fig.5 in the appendix.  We are ready to include a full comparison to DeepR and SET in an additional appendix.  We also publicized source code for these experiments in addition to that required to reproduce all results in the paper (see response above to the DeepSparse team).
> > > Hence, by the above facts, we stand by our claim that "we are the first dynamic sparse reparameterization method to perform on par with or better than pruning-based compression techniques such as Zhu et al. 2017."
> > >
> > > **Claim 2:**
> > > Thank you for acknowledgement of our claim of contribution.  Your criticism is rather on whether our contribution is valuable or trivial, on which any reader of our paper may have a slightly different opinion from the next--not a factual error in our claim.
> > > We believe the demonstration of the ability to scale up dynamic sparse training from a single layer to a deep network, and from toy-sized models to real-world applications, is rather nontrivial.  This is what previous work such as DeepR and SET did not show, and a key consequence of the improved efficiency and scalability achieved by our method.

---

> > > > ### Author Response · Authors · 2018-10-24
> > > > **Authors' response (2/2)**
> > > >
> > > >
> > > > **Claim 4:**
> > > > Our observation of the 5x slowdown of DeepR compared to ours was intended to demonstrate the computational efficiency of our method as well as why we did not include DeepR in the comparison as it is computationally expensive for large dataset/model.
> > > > Though you correctly pointed out two mechanistic differences between our method and SET, we believe a much more consequential difference is downplayed--our method produced sparse models that generalize better.  Just the superior performance should, in our opinion, warrant a closer look at the underlying mechanisms that gave rise to these advantages, however simple these mechanisms might seem.
> > > > In fact, the mechanistic differences between our method and SET are not trivial.
> > > > First, the higher computational efficiency of our method over SET stems from the fact that SET uses a sorting operation over all the weights in a layer whereas ours uses a comparison operation against a threshold for pruning.  This was stated in the manuscript, see second bullet point at the end of page 2 and the third line in page 9.
> > > > Second, automatic parameter reallocation across layers is a key feature of our algorithm that is entirely novel from SET or DeepR and directly contributed to its superior scalability (eliminating the need to configure sparsity for different layers manually) and superior performance (see Appendix C).
> > > > Per your suggestion, we did further experiments using the exact form (i.e. cubic) of the sparsity schedule in Zhu et al. 2017 (tensorflow.contrib.model_pruning).  The difference from our choice (i.e. exponential) is inconsequential.  In the table below we list further experimental results (in test accuracy%) for WRN-28-2 on CIFAR10, for a direct comparison of our method, Zhu et al. 2017, Bellec et al. 2017 (DeepR), and Mocanu et al. 2018 (SET), source code for reproduction also publicized.
> > > > Because the DeepR paper did not provide code or hyperparameter settings for the larger-scale experiments we did, we ran a systematic sweep on the parameters of DeepR (as well as on its temperature annealing schedule) and reported the best results here.
> > > > As you mentioned, demonstrating the superiority of our method over SET and DeepR in terms of accuracy is a valuable contribution. We intend to include the following results as well as further results comparing against SET and DeepR in Imagenet experiments in our paper.
> > > > +-----------------------------------------------------------------------------+
> > > > | Sparsity                                    |         0.9        |         0.8         |
> > > > +-----------------------------------------------------------------------------+
> > > > | Bellec et al. 2017 (DeepR)    | 90.81 ± 0.07 | 91.76 ± 0.22 |
> > > > +-----------------------------------------------------------------------------+
> > > > | Mocanu et al. 2018 (SET)      | 93.42 ± 0.24 | 94.02 ± 0.09 |
> > > > +-----------------------------------------------------------------------------+
> > > > | Zhu et al. 2017                        | 93.76 ± 0.08 | 94.16 ± 0.12 |
> > > > +-----------------------------------------------------------------------------+
> > > > | Ours                                         | 93.68 ± 0.12 | 94.34 ± 0.16 |
> > > > +-----------------------------------------------------------------------------+
> > > >
> > > > Thank you again and we are happy to address your further comments.

---

### Public Comment · (anonymous) · 2018-10-21
**Provide link to code**

Dear Authors,

As part of the of the ICLR reproducibility challenge, our team has selected this paper for replication.
In order to facilitate the process, we  kindly request you send  the link to your code to deepsparse(at)gmail(dot)com.

Looking forward to your response.

---

> ### Public Comment · (anonymous) · 2018-10-22
> **🤔🤔**
>
> suspicious

---

> > ### Public Comment · (anonymous) · 2018-10-22
> > **What's the suspicion**
> >
> > I'm not sure what the suspicion is. The link to our registration is provided [1] and the challenge guidelines say: "If available, the authors' code can and should be used; authors of ICLR submissions are encouraged to release their code to facilitate this challenge." [2]. Also, the authors mention regarding their code (Page 12, Appendix A): "Link suppressed for the sake of anonymity during review process." hence the request for the code.
> >
> > [1] - https://github.com/reproducibility-challenge/iclr_2019/issues/31
> > [2] - https://github.com/reproducibility-challenge/iclr_2019

---

> > > ### Author Response · Authors · 2018-10-24
> > > **Authors' response, link to code, and request for your response**
> > >
> > >
> > > Dear DeepSparse team,
> > >
> > > Thank you for taking the effort to assess our results as part of the 2019 ICLR reproducibility challenge.
> > >
> > > Since there is also an anonymous commenter (see below) raising questions on the details of our technique and its claimed superior performance over previous methods, we choose to publicize the source code for reproducing all results in our paper here in this interactive forum, so that all commenters and reviewers, including you, will be able to validate our results.  Since the paper is still under double-blind review, we set up an anonymous repo to host the code ( https://gitlab.com/anon-dynamic-reparam/iclr2019-dynamic-reparam ).
> > >
> > > Along with the source code, we wish to make the following comments/requests to you:
> > >
> > > (1) Please disclose the academic affiliation of your team, which is a common practice exercised by all other teams of the reproducibility challenge ( https://github.com/reproducibility-challenge/iclr_2019/issues ).
> > >
> > > (2) In spirit of the reproducibility challenge, as instructed on its main page ( https://github.com/reproducibility-challenge/iclr_2019 ): "You are encouraged to contact the authors in private to clarify doubts regarding the paper but you should maintain your anonymity in the issue section before your report submission", please contact us for any questions you might have during your validation.  We would prefer that we communicate in this interactive forum so that (a) anonymity is guaranteed per requirements of the reproducibility challenge, and (b) other commenters/reviewers could also see our communications since our code is shared with all participants of this forum.
> > >
> > > (3) In the repo, you will find YAML files with specific commands and arguments to reproduce the experiments presented in the paper.  In response to specific comments by an earlier anonymous commenter, we also provided extra experiments in a separate YAML file for comparison with earlier methods such as DeepR, which is designed to address questions raised by the commenter, thus not an original part of the manuscript.  Though you are more than welcome to run these experiments as well (our code implements a variety of dynamic reparameterization methods including DeepR and SET; detailed results of our own runs will also be presented in our response to the anonymous commenter below), we would like to caution you that (a) they are not part of the original paper for the reproducibility study, and (b) since earlier methods like DeepR were not published with source code for the large models/datasets we tested here, we did our own implementation based on the original DeepR paper and on discussions with its authors, so in case you have questions on these baseline methods please contact the authors of the original paper for details.
> > >
> > > Thank you again and please kindly reply to our request for your affiliation.
> > >
> > > Kind regards,
> > >
> > > Authors

---

> > > > ### Public Comment · (anonymous) · 2018-10-24
> > > > **Response to affiliations**
> > > >
> > > > Dear Authors,
> > > >
> > > > Thank you for sharing your code and relevant comments.
> > > >
> > > > As for our academic affiliations, we currently do not have any. We regard ourselves as independent research trainees who are enthusiastic about Machine Learning research and are willing to contribute to the field. Moreover, it is stated on the challenge page: "Participation by other researchers or research trainees with adequate machine learning experience is also encouraged" (https://github.com/reproducibility-challenge/iclr_2019).
> > > >
> > > > Regards,
> > > >
> > > > DeepSparse Team

---

> > > > > ### Author Response · Authors · 2018-10-24
> > > > > **Authors response**
> > > > >
> > > > > Thank you for your response. Having no affiliation is unconventional, but in all cases, we are happy to address any questions you have regarding the code or the experiments.
> > > > > Kind regards,
> > > > > Authors

---

### Public Comment · (anonymous) · 2018-10-26
**Appropriate Messaging**

We believe it would be easier to understand the paper’s contribution if the writing style of the paper was less sensationalist.  Rather than focusing on being first to exceed some relatively arbitrary level of performance by a tiny margin and the first to apply this technique to a “modern all convolutional network” (conceptually, nothing prevented any of the previously proposed approaches from being applied to a similar network), the paper could instead make more clear its contributions are two modifications to SET that result in a small improvement in accuracy.  The authors have stated multiple times that their modifications are more efficient, yet they have yet to provide any actual runtime numbers to show that the theoretical improvement matters in practice.

---

> ### Author Response · Authors · 2018-10-28
> **Authors' response (1/2)**
>
>
> Thank you for your comments.  Please see our responses below.
>
> (1) We did not intend to be "sensationalist".  In writing the manuscript, we clearly described our method and objectively stated claims of contributions.  We believe they are factually correct.  We are happy to address any of your specific confusions on our stated contributions that are not "easy to understand".
>
> (2) We agree with you that "nothing prevented any of the previously proposed approaches from being applied to a similar network", but the fact that they did not do so leaves open the question of whether those proposed approaches were applicable to large-scale networks in practice, a question we answered in this work.  We fully recognize the contributions of previous approaches, i.e. DeepR, NeST and SET, and we explicitly stated in our paper that we were inspired by them (see Discussion on Page 8).  As we replied to another anonymous commenter below, our claim of contribution is factually correct, and we believe that our demonstration of the ability to scale up dynamic sparse training from a single layer to a deep network, and from toy-sized models to real-world applications, is rather nontrivial.
>
> (3) The choice of baseline method we benchmarked against (i.e. sparse compression by iterative training and pruning, Zhu et al. 2017) is not arbitrary for the following reasons:
> a) It was the strongest baseline performance known to us.
> b) It, unlike DeepR, SET and ours, is a compression method which does not impose a reduced parameter budget during the entire course, but only at the end, of training.  Matching this baseline has significant implications for direct training of compact sparse deep CNNs without compression--equally effective training can now be done under strict parameter constraints without the need to train a large model first and then followed by compression.  Closing this gap was not achieved by previous methods like DeepR and SET, but by ours in this work.
>
> (4) Further, as we stated in response to the commenter below, the key advancement of our method as compared to SET is its ability to produce sparse models that generalize better, matching the best compression benchmark known to us so far.  To argue against your statement on our improvement "exceed some relatively arbitrary level of performance by a tiny margin", and "two modifications to SET that result in a small improvement in accuracy", we did further statistical analysis (p-values of T-tests) on the extra experiments presented in the table of our reply to the commenter below (WRN-28-2 on CIFAR10), the improvements were highly significant statistically.
> +----------------------------------------------------------------------------------------------+
> | Hypothesis test                                      | Sparsity = 0.9 | Sparsity = 0.8 |
> +----------------------------------------------------------------------------------------------+
> | Ours vs. Bellec et al. 2017 (DeepR)    |    *0.000002    |    *0.000103    |
> +----------------------------------------------------------------------------------------------+
> | Ours vs. Mocanu et al. 2018 (SET)      |     0.420649     |    *0.006790    |
> +----------------------------------------------------------------------------------------------+
> | Ours vs. Zhu et al. 2017                        |     0.163627     |      0.152581    |
> +----------------------------------------------------------------------------------------------+

---

> > ### Author Response · Authors · 2018-10-28
> > **Authors' response (2/2)**
> >
> >
> > This superior accuracy achieved by our method was observed in all networks and benchmarks that we tested:
> > a) MNIST test accuracies (%):
> > +----------------------------------------------------------------------------------------------+
> > | Method                                   |     Sparsity = 0.99    |    Sparsity = 0.98    |
> > +----------------------------------------------------------------------------------------------+
> > | Mocanu et al. 2018 (SET)     |      70.00 ± 13.37      |      97.85 ± 0.11      |
> > +----------------------------------------------------------------------------------------------+
> > | Ours                                        |      97.78 ± 0.098      |     98.08 ± 0.061     |
> > +----------------------------------------------------------------------------------------------+
> >
> > b) We did further Imagenet experiments to address your concerns, here we report single-run test accuracies (% of top-1, top-5) pending the completion of the rest of the 5 runs:
> > +----------------------------------------------------------------------------------------------+
> > | Method                                    |     Sparsity = 0.9     |     Sparsity = 0.8     |
> > +----------------------------------------------------------------------------------------------+
> > | Mocanu et al. 2018 (SET)      |         70.4, 90.1        |         72.6, 91.2        |
> > +----------------------------------------------------------------------------------------------+
> > | Ours                                         |         71.6, 90.5        |         73.3, 92.4        |
> > +----------------------------------------------------------------------------------------------+
> >
> > (5) The two mechanistic differences from SET that gave rise to the significantly better performance of our method were indeed important, and we gave particular attention to them in the manuscript, pointing out that they made our method more performant, more efficient and more scalable.
> >
> > (6) In response to your criticism "they have yet to provide any actual runtime numbers to show that the theoretical improvement matters in practice", here we give specific wall-clock time (in seconds) for reparameterization costs of our method and SET for Resnet-50 on Imagenet (we have already given the anonymous commenter below the cost of DeepR being roughly 5x of ours):
> > +------------------------------------------------------------------------------------------+
> > | Method                                    |     CPU (Xeon)     |   GPU (Titan-XP)   |
> > +------------------------------------------------------------------------------------------+
> > | Mocanu et al. 2018 (SET)      |           0.59            |            0.064           |
> > +------------------------------------------------------------------------------------------+
> > | Ours                                         |           0.29            |            0.045           |
> > +------------------------------------------------------------------------------------------+
> >
> > As you can see, our reparameterization procedure is significantly cheaper than SET.  Remember that, when applied according to the same schedule, our reparameterization method produced significantly better sparse models than SET (see above).
> >
> > As such, we believe we have presented concrete facts to support our claims.
> >
> > Finally, we wish to point out that we have publicized all source code in this interactive forum (see below), and this paper has been selected for the ICLR 2019 reproducibility challenge and is being validated by a third-party team.  We encourage you to validate our results as well and to check the specific claims we make.
> >
> > Thank you again and we are happy to address your further concerns.

---

> > > ### Public Comment · (anonymous) · 2018-10-29
> > > **reply**
> > >
> > > Does an improvement of .02 seconds per step on ImageNet matter?  I think this is a small fraction of the overall step time...  Reporting time relative to the overall step time would be far more useful.  Reporting total training time would be the most useful...
> > >
> > > We will have to agree to disagree about whether to use the term "significant" or "small" for the performance increases which are [.2%, .4%, 1.0%, 1.7%] with the exception of the 99% MNIST experiment.  It was not intended in the sense of statistically "significant"; merely as a description of magnitude.
> > >
> > > If the 99% results are general then _that_ would be a most interesting realm to explore.  I'm guessing this is where the re-parameterization really helps by moving more weights to the final layer?
> > >
> > > We find these improvements interesting and valuable; as we said before we were mostly concerned with the messaging.

---

> > > > ### Author Response · Authors · 2018-10-29
> > > > **Authors' response: thank you for the valuable comments!**
> > > >
> > > >
> > > > We absolutely agree with you: any _quantitative_ improvements in performance compared to previous methods like SET, whether statistically significant or not, are hard to justify significant achievement.  The next method might just improve yet a bit more, so what's special about this one?
> > > > Our argument is rather that our method's achievement is _qualitatively_ significant.  This is manifested in the benchmark against state-of-the-art compression by iterative pruning (Zhu et al. 2017).  Note that this is a compression method which first trains a large, dense network and then compresses it down, but our method and SET both train small sparse network from the very beginning of training.  Even though SET is just slightly worse than our method, SET actually fared significantly (statistically speaking) worse than state-of-the-art compression in at least some cases.  In contrast, even though our method is just slightly better than SET, ours performed at least indistinguishably well or in some cases significantly better (statistically speaking) than state-of-the-art compression; this is true in all experiments we did.  Therefore, the difference is _qualitative_: we fully close this gap, showing for the first time one can train a compact sparse model from scratch to reach at least the same performance of the best known post-training compression method.
> > > >
> > > > We appreciate your criticism of the messaging which will be helpful to us when revising the manuscript. We will be cautious when using the term 'first' as it drew criticism from multiple commenters. We will focus instead on just stating our contributions in a clear and unambiguous manner.
> > > >
> > > > Yes, we agree with your comment on run time.  Both SET and ours, unlike DeepR, perform parameter reallocation rather infrequently during training (once per hundreds of batch iterations), thusly the overhead is relatively negligible despite the fact that our replacement of sorting by comparison makes ours even cheaper than SET.  We wish to point out two things in this response:
> > > > a) Computational cost is a secondary argument in our paper, a much more important advantage of our method over SET is its ability to produce better sparse models, ones that matched state-of-the-art compression.
> > > > b) As the number of parameters increase, sorting scales worse (n*log[n]) than comparison (n).  So in case parameter reallocation has to be done more frequently and in much larger models, there might be a substantial difference in computational overhead.  (This is just another secondary argument that does not make a major point of the paper.)

---

### Meta-Review · Area_Chair1 · 2018-12-14
**Revised draft is a significant improvement over the initial submission**

**Confidence:** 3
**Recommendation:** Reject

**Metareview:**


The authors presents a technique for training neural networks, through dynamic sparse reparameterization. The work builds on previous work notably SET (Mocanu et al., 18), but the authors propose to use an adaptive threshold for and a heuristic for determining how to reparameterize weights across layers.
The reviewers raised a number of concerns on the original manuscript, most notably 1) that the work lacked comparisons against existing dynamic reparameterization schemes, 2) an analysis of the computational complexity of the proposed method relative to other works, and that 3) the work is an incremental improvement over SET.
In the revised version, the authors revised the paper to address the various concerns raised by the reviewers. To address weakness 1) the authors ran experiments comparing the proposed approach to SET and DeepR, and demonstrated that the proposed method performs at least as well, or is better than either approach. While the new draft is in the ACs view a significant improvement over the initial version, the reviewers still had concerns about the fact that the work appears to be incremental relative to SET, and that the differences in performance between the two models were not very large (although the author’s note that the differences are statistically significant). The reviewers were not entirely unanimous in their decision, which meant that the scores that this work received placed it at the borderline for acceptance. As such, the AC ultimately decide to recommend rejection, though the authors are encouraged to resubmit the revised version of the paper to a future venue.